# Histone deacetylase knockouts modify transcription, CAG instability and nuclear pathology in Huntington disease mice

Marina Kovalenko[1], Serkan Erdin[1,2], Marissa A Andrew[1], Jason St Claire[1], Melissa Shaughnessey[1], Leroy Hubert[3], João Luís Neto[1], Alexei Stortchevoi[1], Daniel M Fass[1], Ricardo Mouro Pinto[1,4], Stephen J Haggarty[1,4], John H Wilson[3], Michael E Talkowski[1,2,4], Vanessa C Wheeler[1,4]*

[1]Center for Genomic Medicine, Harvard Medical School, Boston, United States; [2]Program in Medical and Population Genetics, Broad Institute of MIT and Harvard, Cambridge, United States; [3]Verna and Marrs McLean Department of Biochemistry and Molecular Biology, Baylor College of Medicine, Houston, United States; [4]Department of Neurology, Massachusetts General Hospital, Harvard Medical School, Boston, United States

**Abstract** Somatic expansion of the Huntington's disease (HD) CAG repeat drives the rate of a pathogenic process ultimately resulting in neuronal cell death. Although mechanisms of toxicity are poorly delineated, transcriptional dysregulation is a likely contributor. To identify modifiers that act at the level of CAG expansion and/or downstream pathogenic processes, we tested the impact of genetic knockout, in $Htt^{Q111}$ mice, of $Hdac2$ or $Hdac3$ in medium-spiny striatal neurons that exhibit extensive CAG expansion and exquisite disease vulnerability. Both knockouts moderately attenuated CAG expansion, with $Hdac2$ knockout decreasing nuclear huntingtin pathology. $Hdac2$ knockout resulted in a substantial transcriptional response that included modification of transcriptional dysregulation elicited by the $Htt^{Q111}$ allele, likely via mechanisms unrelated to instability suppression. Our results identify novel modifiers of different aspects of HD pathogenesis in medium-spiny neurons and highlight a complex relationship between the expanded $Htt$ allele and $Hdac2$ with implications for targeting transcriptional dysregulation in HD.

*For correspondence:
wheeler@helix.mgh.harvard.edu

## Introduction

Huntington's disease (HD) is a dominantly inherited neurodegenerative disorder typically manifesting in midlife with motor, cognitive, and psychiatric symptoms, leading to death after 15–20 years (*Vonsattel et al., 1985*; *Harper, 1999*). HD is caused by the inheritance of an expansion >35 repeats of a polymorphic CAG repeat tract in the *HTT* gene (*Macdonald et al., 1993*), ultimately resulting in cellular dysfunction and death, with medium-spiny neurons (MSNs) of the striatum being exquisitely sensitive to this mutation (*Vonsattel et al., 1985*). The expanded *HTT* CAG repeat undergoes further time-dependent, CAG length-dependent and tissue/cell-type-dependent expansion (*Wheeler et al., 1999*; *Kennedy and Shelbourne, 2000*; *Kennedy et al., 2003*; *Veitch et al., 2007*; *Gonitel et al., 2008*; *Swami et al., 2009*; *Lee et al., 2010*; *Lee et al., 2011*; *Kovalenko et al., 2012*; *Larson et al., 2015*; *Geraerts et al., 2016*; *Ament et al., 2017*; *Mouro Pinto et al., 2020*). The repeat is highly unstable in the brain, particularly in MSNs (*Kovalenko et al., 2012*), with individual-specific differences in the extent of somatic CAG expansion in HD postmortem brain associated with age of onset (*Swami et al., 2009*). Recent genome-wide association studies (GWAS) for modifiers of HD onset highlight somatic CAG expansion as a key driver of the rate of disease onset (*Genetic Modifiers of Huntington's Disease (GeM-HD)*

*Consortium, 2019*). Genetic data from these GWAS as well as extensive cross-tissue analyses of somatic instability (*Mouro Pinto et al., 2020*) support a two-step model of HD pathogenesis whereby cellular vulnerability is determined by both the rate of somatic CAG expansion and a toxic process(es) triggered by somatically expanded repeats. Thus, a comprehensive understanding of HD pathogenesis will necessitate insight into mechanisms underlying both CAG instability and cellular toxicity.

HD mouse models provide valuable systems in which to identify genetic modifiers. As the two components of the HD pathogenic process outlined above are separable mechanistically, they can be influenced by different modifiers, or by the same modifier via different underlying mechanisms. However, modifiers influencing somatic expansion may also alter downstream phenotypes depending on the sensitivity to detect the impact of altered CAG length. Several DNA repair genes modify somatic CAG expansion in HD mouse models, with mismatch repair (MMR) genes being critical drivers of this process. (*Wheeler et al., 2003*; *Dragileva et al., 2009*; *Tomé et al., 2013*; *Pinto et al., 2013*; *Kovtun et al., 2007*; *Mollersen et al., 2012*). Knowledge of additional modifiers in the mouse is likely to shed insight into underlying mechanisms that are currently unknown, or into ways in which the process might be regulated. For example, local chromatin structure has been suggested to influence repeat instability (*Gorbunova et al., 2004*; *Jung and Bonini, 2007*; *Libby et al., 2003*; *Libby et al., 2008*; *Dion et al., 2008*; *Yang and Freudenreich, 2010*; *Nestor and Monckton, 2011*; *Neto et al., 2017*). The precise molecular underpinnings of cellular toxicity also remain to be defined. However, global transcriptional dysregulation and epigenetic alterations have emerged as likely significant contributors to disease pathogenesis (*Glajch and Sadri-Vakili, 2015*; *Sharma and Taliyan, 2015*). There is an extensive history of investigating histone deacetylase (HDAC) inhibitors in HD and several studies have shown beneficial effects in HD mouse models (*Hockly et al., 2003*; *Gardian et al., 2005*; *Thomas et al., 2008*; *Mielcarek et al., 2011*; *Jia et al., 2012a*; *Jia et al., 2012b*; *Jia et al., 2016*; *Chopra et al., 2016*; *Suelves et al., 2017*; *Siebzehnrübl et al., 2018*). Interestingly one study found that a selective HDAC3 inhibitor also reduced CAG expansion in HD mice (*Suelves et al., 2017*).

Here, with the overarching goal of identifying novel HD modifiers, and with the possibility that epigenetic alterations may contribute both to CAG instability and cellular toxicity processes, we tested the roles of specific histone deacetylases (HDACs) in these different aspects of HD pathogenesis. We have taken a genetic approach using a well-established precise genetic HD knock-in mouse model ($Htt^{Q111}$) (*Wheeler et al., 2000*) and conditionally deleted either *Hdac2* or *Hdac3*, encoding class I HDACs implicated by previous pharmacological studies, specifically in striatal MSNs. We performed RNA sequencing (RNA-seq) to investigate the impact of MSN-specific *Hdac2* knockout on the transcriptome and on gene expression changes elicited by the $Htt^{Q111}$ allele, and tested the effect of MSN-specific *Hdac2* and *Hdac3* knockout on CAG instability and on nuclear huntingtin pathology, both of which occur selectively in MSNs in these mice (*Kovalenko et al., 2012*; *Wheeler et al., 2000*).

## Results

### MSN-specific knockout of *Hdac2* and *Hdac3* in $Htt^{Q111}$ knock-in mice

To investigate the role of HDACs in $Htt^{Q111}$ mice we took a genetic approach, focusing on two Class I HDACs, HDAC2 and HDAC3, due to previous interest in therapeutic targeting of this class of HDACs in HD (*Glajch and Sadri-Vakili, 2015*) and the relatively high expression levels of these HDACs in the striatum and in neurons (*Broide et al., 2007*). As constitutive genetic knockout of *Hdac2* can result in perinatal lethality (*Trivedi et al., 2007*; *Montgomery et al., 2007*), and knockout of *Hdac3* results in early embryonic lethality (*Montgomery et al., 2008*), we took advantage of conditional 'floxed' *Hdac2* and *Hdac3* alleles (*Montgomery et al., 2007*; *Montgomery et al., 2008*), and deleted each of these genes in striatal MSNs of heterozygous $Htt^{Q111/+}$ mice in crosses together with 'DARPP-32 Cre' (D9-Cre) transgenic mice expressing Cre recombinase under the control of the *Ppp1r1b* gene promoter and regulatory elements from 5 to 6 weeks of age (*Bogush et al., 2005*; *Figure 1A*; *Figure 1—figure supplement 1*). We previously used these mice in a similar strategy to delete the *Msh2* gene in MSNs (*Kovalenko et al., 2012*). We generated $Htt^{Q111/+}$ or $Htt^{+/+}$ mice harboring two floxed *Hdac2* or *Hdac3* alleles together with the D9-Cre transgene, resulting in

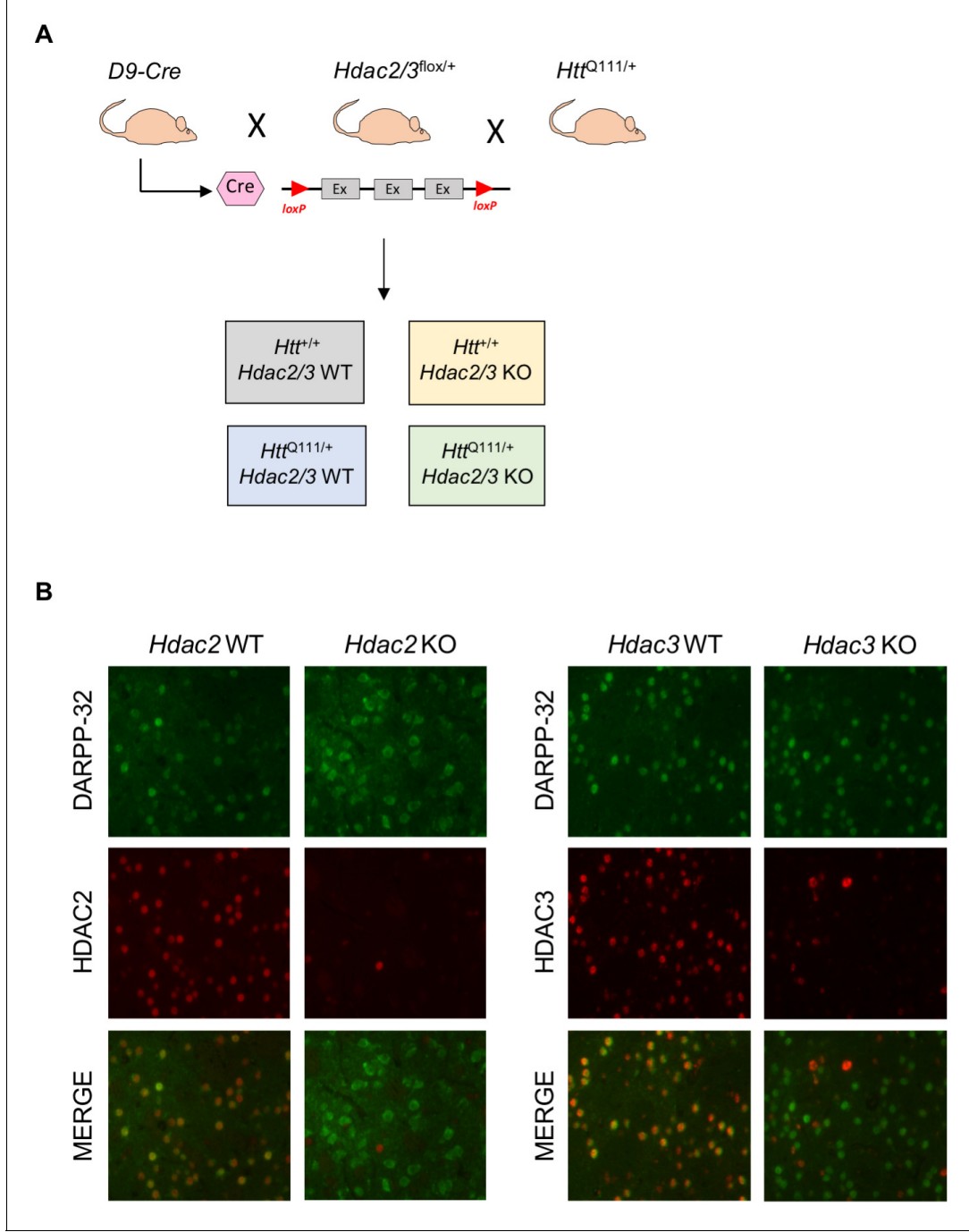

**Figure 1.** MSN-specific deletion of *Hdac2* or *Hdac3*. (**A**) D9-Cre mice, expressing Cre recombinase from the *Ppp1r1b* promoter from 5 to 6 weeks of age, were crossed with floxed *Hdac2* or *Hdac3* mice (schematic illustrates exons flanked by *loxP* sites), and with $Htt^{Q111/+}$ mice to obtain mice with or without a single $Htt^{Q111}$ allele either expressing HDAC2/3 (WT) or with *Hdac2* or *Hdac3* allele deleted in MSNs (KO). See ***Figure 1—figure supplement 1*** for a detailed breeding scheme. (**B**) Fluorescent micrographs of 10 week $Htt^{Q111/+}$ *Hdac2* KO (left) and $Htt^{Q111/+}$ *Hdac3* KO (right) striata co-stained with DARPP-32/HDAC2 and DARPP-32/HDAC3 antibodies, respectively.

The online version of this article includes the following source data and figure supplement(s) for figure 1:

**Figure supplement 1.** Detailed mating scheme.

**Figure supplement 2.** Upregulation of HDAC1 in *Hdac2* KO striata.

**Figure supplement 2—source data 1.** HDAC2 and HDAC1 protein levels in Hdac2 KO mice.

**Figure supplement 3.** *Hdac2* or *Hdac3* knockout in striatal MSNs does not affect HDAC2 or HDAC3 levels in cortex.

deletion of the *Hdac2* or *Hdac3* alleles in MSNs (*Hdac2* or *Hdac3* knockout [KO]) and their litter-mates expressing both *Hdac2* or *Hdac3* alleles (*Hdac2* WT or *Hdac3* WT). HDAC2 or HDAC3 protein are no longer detectable in the majority of the striatal cells in *Hdac2* KO or *Hdac3* KO mice (*Figure 1B*). The remaining HDAC2- or HDAC3-immunopositive cells are DARPP-32 negative, reflecting continued expression in cell types other than MSNs in the striatum (*Figure 1B*). Western blots also confirmed the loss of HDAC2 protein in striata from *Hdac2* KO mice and revealed a compensatory increase in HDAC1 protein levels as seen in other systems with long-term loss of HDAC2 (*Figure 1—figure supplement 2*). As expected, HDAC2 and HDAC3 expression were not altered in the cortex (*Figure 1—figure supplement 3*). Whereas *Hdac2* KO mice appeared indistinguishable from their littermates, *Hdac3* KO mice were significantly smaller in size and had lower body weight, with more than half of the total number of *Hdac3* KO pups not surviving to five months of age, precluding some aspects of this study; notably, we were unable to generate sufficient numbers of *Hdac3* KO mice to perform RNA-seq. We hypothesize that this may be the result of *Ppp1r1b*-driven Cre expression in cell types outside MSNs during development and/or in the periphery that are sensitive to levels of HDAC3, for example in the gut or liver (*Alenghat et al., 2013*; *Knutson et al., 2008*). Nonetheless, this observation highlights the non-redundant functions of HDAC2 and HDAC3 within the mammalian body.

## Impact of MSN-specific *Hdac2* knockout and of the *Htt*^Q111^ allele on the transcriptome

Due to the central role of HDAC2 in regulating transcription (*Kelly and Cowley, 2013*) and previous evidence for a beneficial effect of HDAC inhibitors in HD mouse models, we first performed RNA-seq in the *Hdac2* KO cohort to assess both the global impact of MSN-specific *Hdac2* KO and the extent to which genes dysregulated by the expression of the *Htt*^Q111^ allele might be reversed. Transcriptional dysregulation occurs in *Htt*^Q111/+^ mice by 2–6 months of age (*Ament et al., 2017*;

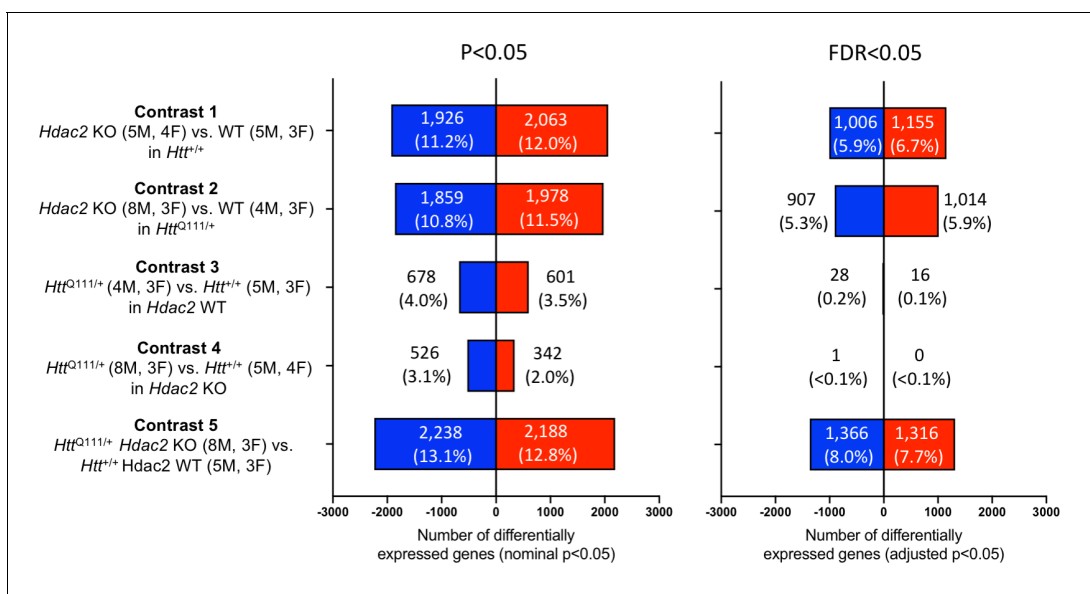

**Figure 2.** RNA-seq analyses in *Htt*^Q111^ striata with MSN-specific *Hdac2* knockout. Results of differential gene expression analyses showing the number of differentially expressed genes (DEGs) for each of five contrasts. Number and % of DEGs at nominal p<0.05 or FDR adjusted p<0.05 are indicated. Total number of genes analyzed: contrast 1 = 17,132; contrast 2 = 17,175; contrast 3 = 17,139; contrast 4 = 17,116; contrast 5 = 17,094. The number of male (M) and female (F) mice in each group is indicated.

The online version of this article includes the following source data and figure supplement(s) for figure 2:

**Figure supplement 1.** Expression of *Hdac* genes and Cre recombinase.

**Figure supplement 2.** Principal Components Analyses (PCA).

**Figure supplement 3.** Global H3K9 and H4K12 acetylation levels.

**Figure supplement 3—source data 1.** AcH3K9 and AcH4K12 levels in Hdac2 KO mice.

**Figure supplement 4.** Cross-study comparison of *Htt*^Q111^-differentially expressed genes.

*Langfelder et al., 2016*; *Bragg et al., 2017*). We performed RNA-seq in striata from 5-month *Htt*^+/+ *Hdac2* WT, *Htt*^Q111/+ *Hdac2* WT, *Htt*^+/+ *Hdac2* KO and *Htt*^Q111/+ *Hdac2* KO mice (*Figure 2*). *Htt*^Q111/+ mice with *Hdac2* WT and *Hdac2* KO genotypes were matched for constitutive CAG repeat lengths (*Source data 1*). Analyses of exon-specific *Hdac2* transcripts revealed marked depletion of reads mapping to exons 2, 3, and 4 as well as the presence of exon 1-exon 5 splice junction reads in the *Hdac2* KO mice, consistent with Cre-mediated deletion of exons 2–4 (*Montgomery et al., 2007*), and further supported by detection of Cre-specific transcripts in these mice (*Figure 2—figure supplement 1*). Analyses of transcript levels of other *Hdac* genes revealed a compensatory increase in *Hdac1* transcripts in *Hdac2* KO striata, consistent with western blots from the same mice (*Figure 1—figure supplement 2*), but no major impact on the expression level of any other *Hdac* gene (*Figure 2—figure supplement 1*). The *Htt*^Q111 allele did not alter expression of *Hdac* genes (*Figure 2—figure supplement 1*), consistent with previous observations that the *HTT* mutation does not influence HDAC expression in primary neurons, or in CAG140 knock-in mice, although expression of some HDACs was altered in R6/2 exon 1 transgenic mice (*Hoshino et al., 2003*; *Quinti et al., 2010*). Principal component analysis (PCA) of the full dataset revealed that the greatest variance between samples was attributable to *Hdac2* genotype, with the second principal component largely separating males and females. There was no clear distinction between samples based on *Htt* genotype (*Figure 2—figure supplement 2*). Global levels of striatal H3K9 and H4K12 acetylation, previously shown to be sensitive to HDAC2 levels in the brain (*Guan et al., 2009*), revealed overall fairly subtle alterations that appeared to depend on age (*Figure 2—figure supplement 3*), consistent with cell-type-specific knockout of *Hdac2* and previous observations that the effects of HDAC inhibition on histone acetylation are restricted to specific genomic loci (*Lopez-Atalaya et al., 2013*).

To assess the impact of the two genetic mutations (*Hdac2* KO and *Htt*^Q111) on the transcriptome we performed differential gene expression analyses to test the effects of the MSN-specific *Hdac2* KO in the *Htt*^+/+ background (contrast 1), or the *Htt*^Q111/+ background (contrast 2), and the effects of the *Htt*^Q111 allele in a *Hdac2* WT background (contrast 3) or the *Hdac2* KO background (contrast 4). We also compared *Htt*^Q111/+ *Hdac2* KO and *Htt*^+/+ *Hdac2* WT striata (contrast 5) (*Figure 2*; *Source data 2*). *Figure 2* displays the number of differentially expressed genes (DEGs) in each of the differential gene expression contrasts. Strikingly, knockout of *Hdac2* in MSNs elicited a substantial transcriptional response, roughly comparable in *Htt*^+/+ or *Htt*^Q111/+ mice, with ~11–12% of genes statistically significantly dysregulated (FDR adjusted $p < 0.05$) and being both up- and down-regulated in approximately equal proportion. In line with previous studies (*Ament et al., 2017*; *Langfelder et al., 2016*; *Bragg et al., 2017*), *Htt*^Q111/+ striata exhibited a large number of genes that were both up- and down-regulated, although only a moderate number reached statistical significance when accounting for multiple testing. Interestingly, the transcriptional dysregulation mediated by the *Htt*^Q111 allele appeared to be somewhat muted in the *Hdac2* knockout background (*Figure 2*-compare contrasts 3 and 4). The comparison of *Htt*^Q111/+ *Hdac2* KO and *Htt*^+/+*Hdac2* WT striata indicated an effect similar in magnitude to that of the *Hdac2* KO mutation alone.

We compared the *Htt*^Q111-DEGs in this study with those in striata of heterozygous *Htt*^Q111/+ mice in a previous study by *Langfelder et al., 2016*. In the Langfelder study, heterozygous *Htt*^Q111/+ mice were analyzed as part of a larger allelic series of *Htt* CAG knock-in mice; here, we reanalyzed the Langfelder data comparing *Htt*^Q111/+ to wild-type striata using the same analysis method used in the current study (*Figure 2—figure supplement 4*; Materials and methods). A large proportion of gene expression changes in the current study overlapped with those in the Langfelder dataset, although many more DEGs were identified in the latter (*Figure 2—figure supplement 4A, B*). Differences may be due to the age of the mice - 5 months in current study and 6 months in the Langfelder study – especially considering the dramatic increase in transcriptional dysregulation that occurs between 2 and 6 months of age (*Langfelder et al., 2016*). In addition, factors such as genetic heterogeneity (Materials and methods) and disproportionate male:female ratios of mice in our study may contribute to reduced power to detect *Htt*^Q111- driven transcriptional changes. Importantly, the vast majority of the common genes were dysregulated in the same direction (*Figure 2—figure supplement 4B*); at a nominal p value of < 0.05, 721 genes were coordinately dysregulated (298 up-regulated in both studies and 433 down-regulated in both studies). Correcting for multiple testing (FDR adjusted $p < 0.05$), 29 genes were coordinately dysregulated (11 up-regulated in both studies and 18 down-regulated in both studies). These 29 genes are displayed in *Figure 2—figure supplement 4C*, highlighting several genes including *Ppp1r1b, Penk and Pde10a* also found to be dysregulated in

$Htt^{Q111/+}$ mice in additional previous studies (*Ament et al., 2017*; *Bragg et al., 2017*). We refer to these 29 genes as 'commonly dysregulated genes' in downstream analyses.

## Overlaps in gene dysregulation elicited by the *Hdac2* KO and the *Htt*^Q111 allele

To gain insight into the transcriptionally dysregulated genes in each of the four contrasts we determined the enrichment of the DEGs in biological pathways (*Figure 3*; *Source data 3*). These analyses revealed that the *Hdac2* KO and *Htt*^Q111 mutations altered a number of shared processes and pathways. For example, pathways related to cell adhesion, plasma membrane, calcium ion binding and morphine addiction were amongst the most significantly up-regulated in response to either the *Hdac2* KO mutation (*Figure 3*, contrast 1) or the *Htt*^Q111 allele (*Figure 3*, contrast 3). Broadly, pathways related to neuronal function were down-regulated in both in *Hdac2* KO striata and $Htt^{Q111/+}$ striata, although the nature of these pathways was somewhat different, for example an enrichment for signaling pathways in $Htt^{Q111/+}$ striata and for pathways related to neuronal and synapse structure in the *Hdac2* KO. To gain further insight into commonalities between the impacts of *Hdac2* KO and *Htt*^Q111 mutations on transcriptional dysregulation we determined the overlaps in the DEGs and in the pathways enriched for DEGs elicited by the two mutations (i.e. overlap between contrasts 1 and 3) (*Figure 3—figure supplement 1*, yellow highlighted cells). There was a highly significant overlap in the DEGs and enriched pathways due to the *Hdac2* KO or the *Htt*^Q111 allele, largely reflecting overlaps in DEGs and enriched pathways altered in the same direction, i.e. up- or down-regulated in both conditions (statistical tests of enrichment were performed using one-sided Fisher exact tests – see Materials and methods, and results are shown in *Figure 3—figure supplement 1C*). We also compared the effect of the *Hdac2* KO in the presence and absence of the *Htt*^Q111 allele (*Figure 3*, contrasts 1 and 2). There were highly significant overlaps in the DEGs and enriched pathways altered in the same direction between two contrasts (*Figure 3—figure supplement 1*; pink highlighted cells). Finally, we compared the effect of *the Htt*^Q111 allele in the presence and absence of the *Hdac2* KO mutation (*Figure 3*, contrasts 3 and 4). The pathways downregulated by the *Htt*^Q111 allele in the *Hdac2* KO mice again largely encompassed those involved in neuronal function, whilst the most significantly upregulated pathways highlighted the extracellular matrix rather than cell adhesion pathways that were predominantly upregulated by the *Htt*^Q111 allele on the *Hdac2* WT background. Again, there were significant overlaps in DEGs and enriched pathways that were altered in the same direction by the *Htt*^Q111 allele in *Hdac2* WT and KO conditions (*Figure 3—figure supplement 1*, blue highlighted cells).

It should be noted that a direct comparison of gene dysregulation elicited by the *Htt*^Q111 allele and *Hdac2* KO is complicated by the fact that the *Htt*^Q111 allele is constitutively expressed whereas the *Hdac2* KO mutation is specific to striatal MSNs. Nevertheless, taken together, the results of our differential gene expression analyses, pathway enrichment and overlap analyses indicate that: (1) Genes and pathways can be dysregulated in the same direction by either the *Htt*^Q111 allele or the *Hdac2* KO mutation; (2) There is substantial overlap in gene dysregulation by the *Hdac2* KO mutation in the presence or absence of the *Htt*^Q111 allele, and vice versa. (3) The impact of one mutation depends in part on the presence of the other.

## A subset of genes dysregulated in *Htt*^Q111/+ striata is reversed by knockout of *Hdac2* in MSNs

We were interested to determine the extent to which *Htt*^Q111-dysregulated genes might be reversed by the MSN-specific *Hdac2* KO. However, given the overlap in gene dysregulation elicited by either the *Htt*^Q111 allele or the *Hdac2* KO (*Figure 3*; *Figure 3—figure supplement 1*) we wondered whether *Htt*^Q111-dysregulated genes might in fact be further changed in the same direction by the *Hdac2* KO. To assess this, we first determined the set of genes that was dysregulated both in the comparison of $Htt^{Q111/+}$ vs. $Htt^{+/+}$ striata in the *Hdac2* WT background (contrast 3) and in the comparison of *Hdac2* KO vs. *Hdac2* WT striata in the $Htt^{Q111/+}$ background (contrast 2) (*Figure 4A*, left table; *Figure 3—figure supplement 1*, green highlighted cells, *Source data 4*). We identified 520 DEGs common to both comparisons applying a nominal p value < 0.05 to each differential gene expression analysis, while 17 were common to both comparisons applying an FDR-adjusted p value<0.05. The overlapping genes included both those that changed in opposite directions, i.e. up-

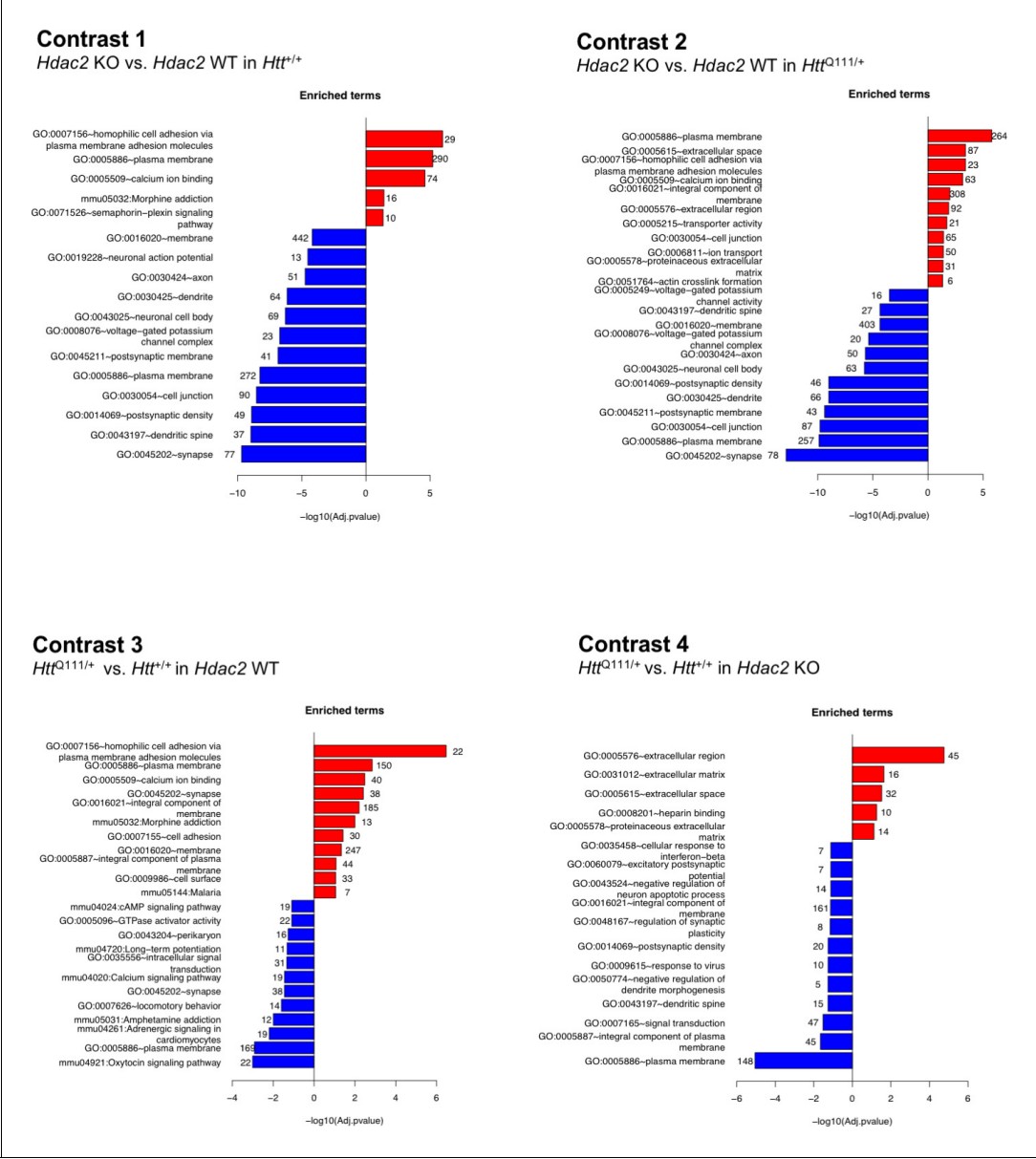

**Figure 3.** Pathway enrichment of differentially expressed genes. The most significantly enriched pathways that are up- or down-regulated in each of four differential gene expression contrasts are shown. Numbers at the end of each bar show the number of genes represented in each pathway. For contrasts 1 and 2, DEGs used in the pathway analyses met the FDR < 0.05 threshold; for contrasts 3 and 4 DEG genes had a nominal p<0.05. The full set of pathways analyses is provided in *Source data 2*.

The online version of this article includes the following figure supplement(s) for figure 3:

**Figure supplement 1.** Overlaps of differentially expressed genes and enriched pathways in each of the four contrasts.

regulated by $Htt^{Q111}$ and then down-regulated by *Hdac2* KO and vice versa, and those that changed in the same direction. The former (69 up-regulated by $Htt^{Q111}$ and down-regulated by *Hdac2* KO plus 144 down-regulated by $Htt^{Q111}$ and up-regulated by *Hdac2* KO) comprise 17% of all $Htt^{Q111}$-dysregulated genes (p<0.05), indicating possible rescue. The latter (158 up-regulated and 149 down-regulated by $Htt^{Q111}$ and by *Hdac2* KO in $Htt^{Q111}$ striata) comprise ~24% of all $Htt^{Q111}$-dysregulated genes (p<0.05), indicating a sizeable proportion of genes for which the $Htt^{Q111}$-mediated dysregulation is exacerbated by the *Hdac2* KO. Therefore, it appears that gene dysregulation elicited by the $Htt^{Q111}$ allele can be both exacerbated and reversed by *Hdac2* KO.

**Figure 4.** Rescue by *Hdac2* KO of gene expression levels of a subset of genes dysregulated by the *Htt*[Q111] allele. (**A**) To identify genes dysregulated by the *Htt*[Q111] allele and by *Hdac2* KO in *Htt*[Q111/+] mice, the overlapping genes in contrasts 3 and 2 were identified (left table). This shows the numbers of genes (p<0.05) up-regulated by *Htt*[Q111] and either further up-regulated, or down-regulated by *Hdac2* KO in *Htt*[Q111/+] mice, and the numbers of genes down-regulated by *Htt*[Q111] and either further down-regulated, or up-regulated by *Hdac2* KO in *Htt*[Q111/+] mice. To identify *Htt*[Q111]-dysregulated genes whose expression level was normalized by *Hdac2* KO, genes from the contrast 3/2 overlap whose expression level did not differ significantly (p>0.05) between *Htt*[+/+] *Hdac2* WT and *Htt*[Q111/+] *Hdac2* KO striata were identified (right table). (**B**) Heat map of the 97 genes down-regulated by *Htt*[Q111] and up-regulated by *Hdac2* KO in *Htt*[Q111/+] striata and of the 55 genes up-regulated by *Htt*[Q111] and down-regulated by *Hdac2* KO in *Htt*[Q111/+] striata,

*Figure 4 continued on next page*

*Figure 4 continued*

whose expression levels did not differ significantly between *Htt*$^{+/+}$ *Hdac2* WT and *Htt*$^{Q111/+}$ *Hdac2* KO. (**C**) Examples of four such genes, two down-regulated by *Htt*$^{Q111}$ and two up-regulated by *Htt*$^{Q111}$ are displayed as box-plots.

The online version of this article includes the following figure supplement(s) for figure 4:

**Figure supplement 1.** Impact of *Hdac2* KO on the most significantly dysregulated genes common to this study and Langfelder et al.

To investigate reversal of *Htt*$^{Q111}$-mediated dysregulation by *Hdac2* KO we performed t-tests on the 520 overlapping genes to define genes whose expression level did not differ significantly (p>0.05) between *Htt*$^{+/+}$ *Hdac2* WT and *Htt*$^{Q111/+}$ *Hdac2* KO striata. i.e. reversion of gene expression to wild-type in the presence of both mutant alleles (*Figure 4A*, right table). 152 genes had expression levels that did not differ significantly between these two genotypes, of which 97 were down-regulated by *Htt*$^{Q111}$ and up-regulated by *Hdac2* KO and 55 were up-regulated by *Htt*$^{Q111}$ and down-regulated by *Hdac2* KO (*Figure 4B*, lower table, *Source data 4*). These genes represent ~12% of the total number of *Htt*$^{Q111}$-dysregulated genes (p<0.05). Note that a far greater number of genes did show significant differences in expression between *Htt*$^{+/+}$ *Hdac2* WT and *Htt*$^{Q111/+}$ *Hdac2* KO striata (*Figure 2*, contrast 5). Gene expression values for these 152 genes across all four genotypes are displayed as a heat map in *Figure 4B*, highlighting the comparable gene expression levels in *Htt*$^{+/+}$ *Hdac2* WT and *Htt*$^{Q111/+}$ *Hdac2* KO striata (middle two panels) for both *Htt*$^{Q111/+}$ down-regulated genes (left hand heat map) and *Htt*$^{Q111/+}$ up-regulated genes (right hand heat map). Also apparent are effects of the *Hdac2* KO on these genes in *Htt*$^{+/+}$ striata (compare second and fourth panels), though to a lesser degree than in *Htt*$^{Q111/+}$ striata. Examples of two of these *Htt*$^{Q111}$ down-regulated genes and two *Htt*$^{Q111}$-up-regulted genes are displayed as box-plots in *Figure 4C*. Of interest, one of these genes is *Tcerg*, a candidate human onset modifier (*Genetic Modifiers of Huntington's Disease (GeM-HD) Consortium, 2019*).

We further specifically examined the impact of *Hdac2* KO on the 29 genes (*Figure 2—figure supplement 4*) commonly dysregulated (FDR adjusted p<0.05) in the current study and in the Langfelder study (*Figure 4—figure supplement 1*) Two of these 29 genes (*Ppp1r1b* and *Slc29a1*) overlapped with the set of 152 genes whose expression level did not differ between *Htt*$^{+/+}$ *Hdac2* WT and *Htt*$^{Q111/+}$ *Hdac2* KO (*Figure 4D*; *Figure 4—figure supplement 1A*). An additional 13 (*Cd200, Dmkn, Dsg, Gpd1, Inpp5j, Myo5c, Pdp1, Ppp4r4, Ptprh, Ptprv, Sbsn, Syp, and Vill)* overlapped with the set of 520 genes whose expression level was altered by *Htt*$^{Q111}$ and by *Hdac2* KO in *Htt*$^{Q111}$ mice, but for which the *Htt*$^{Q111}$-mediated gene dysregulation was not normalized by *Hdac2* KO in *Htt*$^{Q111}$ mice. For the majority of these genes, *Hdac2* KO altered gene expression in the same direction as the *Htt*$^{Q111}$ allele, indicating an exacerbation of the *Htt*$^{Q111}$-mediated dysregulation (*Figure 4—figure supplement 1B*). For 14 of the 29 commonly *Htt*$^{Q111}$-dysregulated genes (*B3gnt2, C4a, Foxp4, Gpr6, Homer1, Lrrtm4, Myadm, Ndst3, Olfm3, Pde10a, Penk, Rgs9, Trpv2, Vrk1*), their expression level was not further altered by the *Hdac2* KO (*Figure 4—figure supplement 1C*).

Interestingly, we noticed that in these 29 commonly dysregulated genes, the *relative* impact of the *Htt*$^{Q111}$ allele was reduced in *Hdac2* KO mice. i.e. the difference in gene expression between *Htt*$^{Q111/+}$ and *Htt*$^{+/+}$ is less in *Hdac2* KO than in *Hdac2* WT striata (*Figure 4—figure supplement 1*). To better visualize this, we plotted the absolute log 2-fold change (i.e. log2 fold-change regardless of the direction of change) between *Htt*$^{Q111/+}$ and *Htt*$^{+/+}$ for each of these genes in *Hdac2* WT mice and in *Hdac2* KO mice (*Figure 5*), highlighting the reduced fold-change for each of these genes in the *Hdac2* KO, though to different degrees. Permutation (Materials and methods) showed a statistically significant enrichment for a reduced relative impact (absolute log2 fold-change) of the *Htt*$^{Q111}$ allele in the *Hdac2* KO in these 29 genes (p<1e-5).

We therefore observe the following: (1) The absolute expression levels of a subset of *Htt*$^{Q111}$-dysregulated genes are effectively normalized in *Htt*$^{Q111}$ mice harboring the *Hdac2* KO mutation to levels present in wild-type mice, suggesting rescue of these gene expression changes; (2) The normalization of absolute expression levels of a subset of *Htt*$^{Q111}$-dysregulated genes occurs in a substantially larger background of gene dysregulation elicited by the *Hdac2* KO; (3) In addition, and apparent in a set of the most significantly *Htt*$^{Q111}$-dysregulated genes, *Hdac2* KO lessens the *relative* impact of the *Htt*$^{Q111}$ mutation, regardless of the absolute effect of the *Hdac2* KO on the level

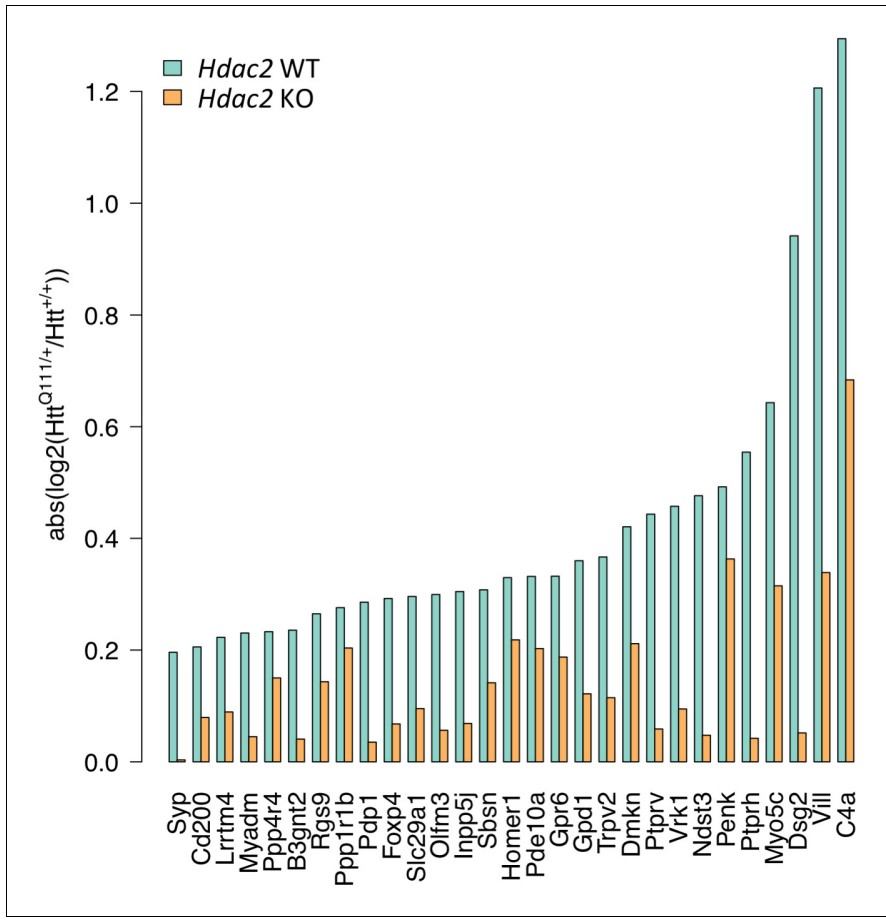

**Figure 5.** *Hdac2* KO reduces the relative impact of the *Htt*[Q111] allele. Absolute log2 fold-change of gene expression differences between *Htt*[Q111/+] and *Htt*[+/+] were determined in *Hdac2* WT mice and in *Hdac2* KO mice as a measure of gene expression change regardless of direction of effect. x-axis indicates the set of 29 genes significantly dysregulated by *Htt*[Q111] (FDR adjusted p<0.05) in both this study and in *Langfelder et al., 2016*; *Figure 2—figure supplement 4*; *Figure 4—figure supplement 1*.

of gene expression. Taken together, our results indicate a potentially complex relationship between *Hdac2* KO and the *Htt*[Q111] mutation.

## *Hdac2* and *Hdac3* modify CAG expansion in *Htt*[Q111] knock-in mice

We next tested the impact of MSN-specific *Hdac2* or *Hdac3* KO on CAG expansion to determine whether these genes might be involved in the mechanism that drives the rate of phenotypic onset in this vulnerable cell population. We analyzed and quantified CAG striatal expansions in ABI Gene-Mapper traces of *HTT* CAG repeat-containing PCR products from cohorts of *Htt*[Q111/+] *Hdac2* KO mice at 2.5, 5, and 10 months of age and from *Htt*[Q111/+] *Hdac3* KO mice at 5 months of age (*Figure 6*). This analysis provides a measure of repeat length distribution across the entire population of striatal cells, and as MSNs contribute to the majority of the unstable alleles at the ages analyzed, it is sensitive to changes in repeat length that are confined to MSNs as shown previously (*Kovalenko et al., 2012*). Shown are representative raw GeneMapper traces (*Figure 6A*) and normalized GeneMapper data, averaged per genotype, showing the height of each peak and the change in CAG length relative to those of the main (modal) allele (*Figure 6B*). We determined an expansion index (*Lee et al., 2010*) from the GeneMapper data, a measure incorporating both the relative height and distance from the main allele of the expansion peaks representing the mean repeat expansion in the cell population (*Figure 6C*). Mean CAG lengths were matched between genotypes for each cohort (*Source data 1*). At 2.5 months of age, we observed a subtle decrease in expansion index in *Hdac2* KO striata that was not statistically significant (*Hdac2* WT: mean 5.76, 95%

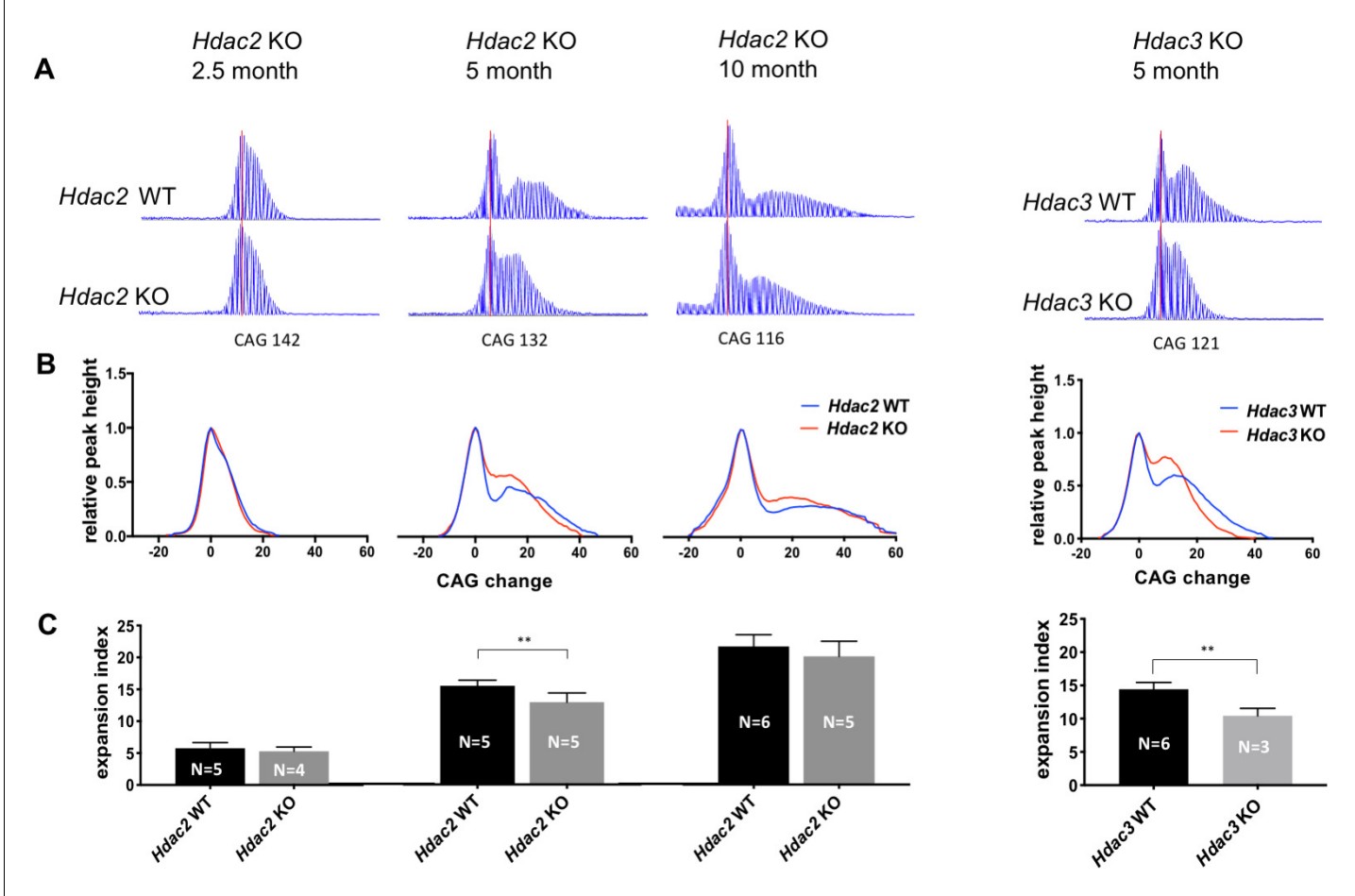

**Figure 6.** Deletion of *Hdac2* or *Hdac3* in striatal MSNs decreases striatal *HTT* CAG repeat expansions. (**A**) Representative GeneMapper traces of PCR-amplified striatal *HTT* CAG repeats from individual mice matched by main allele repeat length within each age group (***Source data 1***). Main allele is marked with a vertical red line and the number of CAG repeats is indicated below. Peaks to the right of the main allele represent CAG repeat expansions. (**B**) Genotype-averaged GeneMapper-derived data showing peak heights normalized by the main allele peak height and CAG change relative to the main allele. (**C**) Mean striatal expansion indices per genotype. Error bars represent standard deviation. **p<0.01 (two-tailed unpaired t-test). *p<0.05, **p<0.01; ***p<0.005 (2-tailed unpaired t-test). Numbers of mice for each group are indicated on the bar graphs.

The online version of this article includes the following source data and figure supplement(s) for figure 6:

**Source data 1.** Expansion Indices in Hdac2 and Hdac3 KO mice.

**Figure supplement 1.** Expression of *Cre* transgene does not affect *HTT* CAG repeat instability and nuclear huntingtin accumulation in MSNs.

**Figure supplement 1—source data 1.** Expansion Indices and mAb5374 nuclear huntingtin immunostaining intensities in Cre-expressing HttQ111 mice.

**Figure supplement 2.** Effect of *Hdac2* or *Hdac3* deletion in striatal MSNs on number of DARPP-32-positive cells and DARPP-32 levels.

**Figure supplement 2—source data 1.** DARPP32-positive cells and DARP32 levels in Hdac2 and Hdac3 KO mice.

**Figure supplement 3.** DNA repair gene expression.

**Figure supplement 3—source data 1.** MSH3 protein levels in Hdac2 KO mice.

CI [4.64, 6.88]; *Hdac2* KO: mean 5.29, 95% CI [4.23,6.35]; two-tailed unpaired t-test p=0.42). At 5 months of age the expansion index was significantly reduced in *Hdac2* KO striata (*Hdac2* WT: mean 15.22, 95% CI [14.16, 16.28]; *Hdac2* KO: mean 12.69, 95% CI [10.91, 14.47]; 2-tailed unpaired t-test p=0.0094), while at 10 months of age, we again observed a small reduction in expansion in *Hdac2* KO striata; this was most apparent from the GeneMapper traces of closely CAG repeat-matched mice (panel A), though this did not reach statistical significance (*Hdac2* WT: mean 21.55, 95% CI [19.62–23.48]; *Hdac2* KO: mean 20.02, 95% CI [17.1–22.94]; two-tailed unpaired t-test p=0.26) across this cohort of mice with a wide range of CAG lengths (***Source data 1***). Knockout of *Hdac3* in MSNs also decreased the expansion index in the striata of 5 month mice (*Hdac3* WT: mean

13.94, 95% CI [12.81–15.07]; *Hdac3* KO: mean 10.43, 95% CI [7.62–13.24]; two-tailed unpaired t-test p=0.0026).

Note that the expression of Cre recombinase itself did not impact CAG expansion (*Figure 6—figure supplement 1*); therefore, the observed impacts on striatal instability are attributable to the *Hdac2* or *Hdac3* genotype of the mice. Further, as neither *Hdac2* KO nor *Hdac3* KO resulted in any significant loss of MSNs as determined by the number of DARPP-32-positive cells (*Figure 6—figure supplement 2*), reduced striatal expansions cannot be readily explained by an altered cellular population biased toward stable alleles. Of note, HDAC3 is required for the long-term maintenance of cerebellar purkinje cells (*Venkatraman et al., 2014*), although its knock-out in the CA1 region of the hippocampus did not have a detrimental effect on cell survival (*McQuown et al., 2011*) in line with the lack of obvious MSN degeneration in our *Hdac3* knockout mice. Thus, HDAC3's role in neuronal survival may depend on cell-type and/or stage of neuronal development or differentiation.

Overall, our results support the conclusion that *Hdac2* and *Hdac3* are moderate enhancers of CAG expansion in MSNs of $Htt^{Q111/+}$ mice, with the impact of *Hdac2* KO appearing to depend on the age of the mice. To explore a possible mechanism for instability modification, we investigated the RNA-seq data for expression levels of DNA repair genes that modify repeat instability in mice (*Kovalenko et al., 2012*; *Dragileva et al., 2009*; *Pinto et al., 2013*; *Kovtun et al., 2007*; *Mollersen et al., 2012*; *Zhao and Usdin, 2018*; *Miller et al., 2020*). Of these, only *Msh3* mRNA levels were modestly decreased in $Htt^{Q111/+}$ *Hdac2* KO striata relative to $Htt^{Q111/+}$ *Hdac2* KO striata (nominal p<0.01; adjusted p=0.07) (*Figure 6—figure supplement 3A, B*). However, MSH3 protein levels were not reduced in *Hdac2* KO striata from these mice (*Figure 6—figure supplement 3C, D*). Therefore, dysregulation of DNA repair gene expression does not provide an obvious explanation for the impact of *Hdac2* KO on CAG expansion.

## Pharmacological HDAC inhibition promotes CAG contraction in a cell-based assay

Previous studies have found that Class I HDAC inhibitors suppress CAG expansion (*Jung and Bonini, 2007*; *Suelves et al., 2017*; *Debacker et al., 2012*). To test further the impact of HDAC inhibition on CAG instability, we took advantage of a well-established, sensitive human cell-based assay for repeat contractions (*Figure 7A*). In this system, a CAG95 repeat is inserted in an intron of an *HPRT* mini-gene construct. Contractions below a threshold of ~38–41 CAGs restore HPRT activity allowing cell growth in selective media. Although the system is blind to expansions, there is relatively good consistency between genetic modifiers identified in this assay and those that modify CAG expansion in mouse models, although the direction of effect can be different (*Lin and Wilson, 2009*). In further support of a role for HDACs as modifiers of CAG instability we found that the small molecule HDAC inhibitors (*Bradner et al., 2010*), SAHA (suberanilohydroxamic acid), and MS-275, both of which share in common the potential inhibition of $Zn^{2+}$-dependent Class I HDACs, and splitomicin, a weak inhibitor of $NAD^+$ dependent Class III HDACs (Sirtuins) (*Carafa et al., 2011*), stimulated CAG contractions (*Figure 7B,C*). Remarkably, we note that the magnitude of effect of the HDAC inhibitors in this contraction assay was comparable to or exceeded that of the various other genetic and pharmacological manipulations we have previously performed (*Lin and Wilson, 2009*; *Lin and Wilson, 2007*; *Lin and Wilson, 2012*; *Hubert et al., 2011*). Note that the enhancement of contractions by HDAC inhibitors in this system is in contrast to the suppressive effect on CAG expansion of genetic or pharmacological inhibition of HDACs in the current study and in previous studies (*Jung and Bonini, 2007*; *Suelves et al., 2017*; *Debacker et al., 2012*), although we cannot exclude the possibility that *Hdac2* or *Hdac3* KO in MSNs also stimulated rare contraction events that are not detectable in the GeneMapper assay. Taken together, the data in mice and in cell models, both in this study and previous studies, suggest that Class I HDACs such as HDAC2 and HDAC3, as well as potentially other HDACs, can influence different aspects of CAG instability, perhaps depending on the cell-type and whether the cell is actively dividing or post-mitotic.

## Impact of MSN-specific *Hdac2* and *Hdac3* knockout on nuclear huntingtin pathology

$Htt^{Q111}$ mice exhibit both early (~2.5 months) diffusely immunostaining nuclear mutant huntingtin, detected with the anti-huntingtin antibody EM48 (or mAb5374), followed at ~6–12 months by the

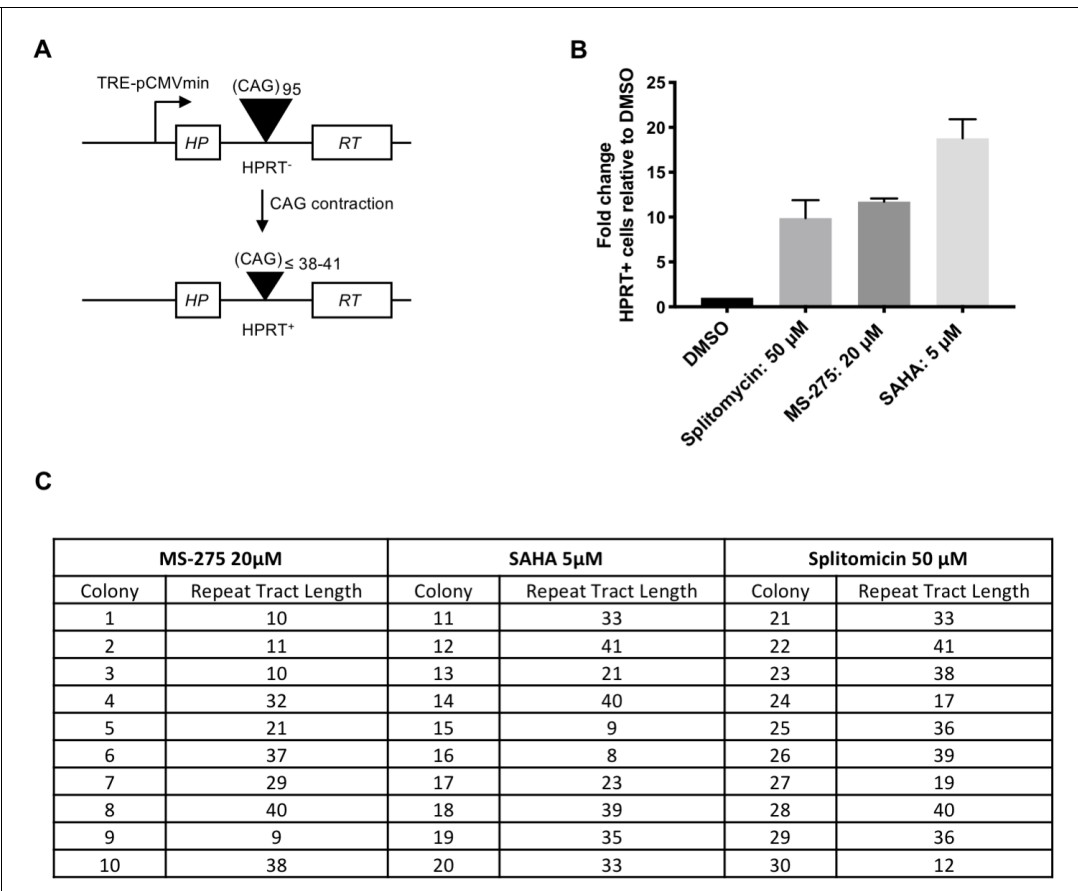

**Figure 7.** HDAC inhibitors increase instability in selectable cell-based assays for CAG repeat contraction. (**A**) *HPRT* mini-gene reporter construct for contraction assay. (**B**) Fold-change in the number of HAT-resistant FLAH25 cells (HPRT+) when treated with HDAC inhibitors relative inhibitors relative to the number of resistant colonies in DMSO-treated cells. Bars show mean +/- SD of bars fold-change relative to DMSO in four biological replicates (independent wells treated with drug). (**C**) CAG repeat lengths in HPRT+ colonies following HDAC inhibitor treatment. DNA was extracted from HPRT+ colonies, and the repeat tract PCR-amplified and sequenced to determine repeat length.

appearance of discreet intra-nuclear inclusions (*Wheeler et al., 2000*; *Wheeler et al., 2002*; *Lloret et al., 2006*). These cellular phenotypes, which occur selectively in MSNs in *Htt*^Q111 mice, are highly CAG repeat length-dependent and are modified by genes that alter somatic expansion (*Kovalenko et al., 2012*; *Pinto et al., 2013*). To assess the impact of MSN-specific *Hdac2* and *Hdac3* KO on these phenotypes we immunostained striatal sections from cohorts of mice (*Hdac2* KO at 2.5, 5 and 10 months of age; *Hdac3* KO at 5 months of age) matched for constitutive CAG repeat length (*Source data 1*) with mAb5374 (*Figure 8*). With the exception of the 2.5-month *Hdac2* KO cohort, mAb5374 immunostaining was performed on the contralateral striata of the same mice analyzed for somatic CAG expansion above.

In 2.5- and 5-month mice, we quantified the intensity of the diffuse nuclear signal. *Hdac2* KO in MSNs reduced the intensity of diffusely immunostaining nuclear mutant huntingtin at both 2.5 months (*Hdac2* WT: mean 153.9, 95% CI [127.9, 179.9]; *Hdac2* KO: mean 96.35, 95% CI [5.63, 187.1]; two-tailed unpaired t-test p=0.025) and 5 months of age (*Hdac2* WT: mean 226.8, 95% CI [159.9, 293.8]; *Hdac2* KO: mean 120.9, 95% CI [77.34, 164.5]; two-tailed unpaired t-test p=0.0062), with a stronger impact at 5 months (*Figure 8A, B*). Diffuse nuclear huntingtin immunostaining was not altered by Cre expression alone (*Figure 6—figure supplement 1*). In 10-month mice, nuclei contained distinct inclusions. As the immunoflourescent signal is saturated in these dense inclusions we quantified their number rather than their intensity. This phenotype was variable between mice of the same genotype, largely due to constitutive CAG length. We therefore separated the mice into 'high' and 'low' CAG groups that reflected different breeding cohorts for comparison between *Hdac2*

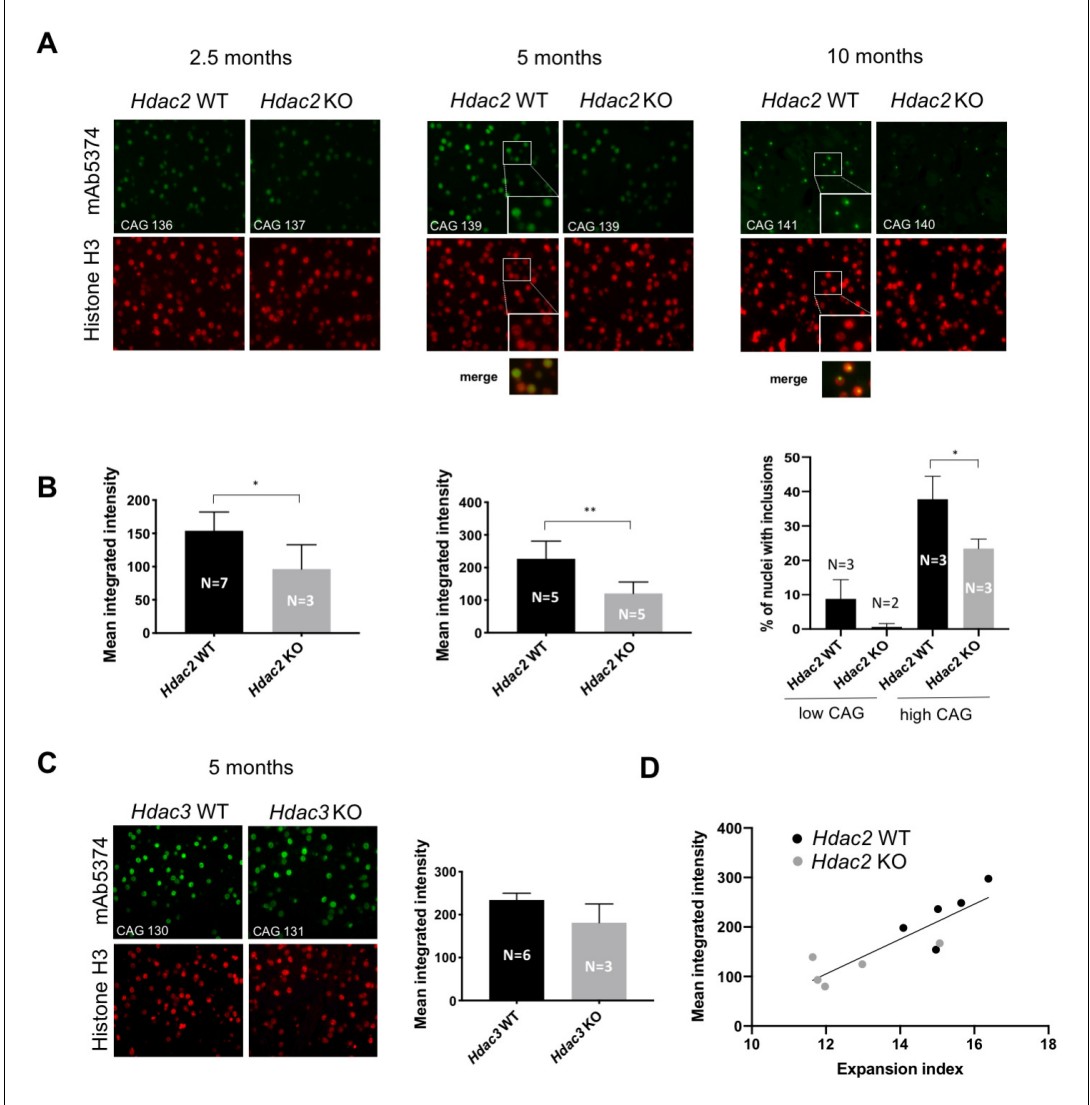

**Figure 8.** Impact of MSN-specific *Hdac2* or *Hdac3* knockout on striatal nuclear mutant huntingtin immunostaining phenotypes. (**A**) Fluorescent micrographs of *Htt*$^{Q111/+}$ *Hdac2* KO striata immunostained with anti-huntingtin mAb5374 and histone H3 antibodies. Mice were matched by CAG repeat length within each age group. Images of 2.5-month and 5-month striata were taken at 40x; images of 10-month striata taken at 20x with 2x digital zoom. The inserts in the 5-month and 10-month *Hdac2* WT mAb5374 images and merged images below highlight the overlap of nuclear huntingtin immunostaining over the entire H3-positive nucleus in the 5-month mice, in contrast to the discrete nuclear huntingtin inclusion immunostaining intensity within the nucleus at 10 months. (**B**) Mean integrated intensity of mAb5374 staining (integrated intensity normalized by the total number of nuclei as determined by the number of H3-positive nuclei) is shown for 2.5-month and 5-month cohorts of *Htt*$^{Q111/+}$ *Hdac2* KO mice. For 10-month mice, the number of nuclei containing an inclusion (we have not observed more than one inclusion per nucleus) was counted as a percentage of the total number of nuclei, determined by histone H3 immunostaining. The 10-month group of mice was divided into two subgroups based on CAG repeat length: low CAG repeat subgroup represented by the first two columns in the graph (*Hdac2* WT CAG 117, 119, 120; *Hdac2* KO CAG 110, 116) and high CAG repeat subgroup represented by the last two columns in the graph (*Hdac2* WT CAG 138, 141, 146; *Hdac2* KO CAG 132, 140, 140). (**C**) Fluorescent micrographs (20x) of 5 month *Hdac3* KO from CAG length-matched mice stained with mAb5374 and histone H3 antibodies (left) and quantified mean integrated intensity of mAb5374 staining (right). 5-months. Numbers of mice for each group are indicated on the bar graphs. The online version of this article includes the following source data and figure supplement(s) for figure 8:

**Source data 1.** mAb5474 nulcear intensity and number of nuclear inclusions in Hdac2 and Hdaac3 KO mice.

**Figure supplement 1.** Huntingtin expression.

**Figure supplement 1—source data 1.** Zip file: RNAseq_codes_and_files.

genotypes (*Figure 8A, B*). We observed a decrease in the number of nuclei containing inclusions in

the low CAG group that did not reach statistical significance (*Hdac2* WT: mean 8.76, 95% CI [−5.29, 22.81]; *Hdac2* KO: mean 0.66, 95% CI [−7.74, 9.06]; 2-tailed unpaired t-test p=0.153), and a statistically significant decrease in the number of cells containing inclusions in the high CAG group (*Hdac2* WT: mean 37.8, 95% CI [21.26, 54.31]; *Hdac2* KO: mean 23.44, 95% CI [16.58, 30.31]; 2-tailed unpaired t-test p=0.0261). In the 5-month *Hdac3* KO cohort, we observed a non-statistically significant trend towards decreased diffuse nuclear mutant huntingtin in a small number of immunostained 5-month *Hdac3* KO striata (*Hdac3* WT: mean 234, 95% CI [91.13, 376.9]; *Hdac3* KO: mean 181.2, 95% CI [72.92, 289.5]; two-tailed unpaired t-test p=0.21). (*Figure 8C*). To examine the relationship between CAG expansion and nuclear huntingtin pathology, we compared directly expansion indices and mAb5374 intensity in the 5-month *Hdac2* KO cohort for which we had both measures from the same mice with a reasonable number (N = 5) of each genotype. There was a strong correlation (Pearson r = 0.87, p=0.0011) between mAb5374 intensity and expansion index (*Figure 8D*), consistent with somatic CAG expansion as a modifier of this cellular phenotype. Overall, the fairly moderate suppression of the mAb5374 nuclear intensity and nuclear inclusion phenotypes appears in line with the subtly reduced somatic expansion elicited by the *Hdac2* or *Hdac3* gene knockouts.

As altered huntingtin levels could contribute to the reduced nuclear immunostaining phenotypes, we investigated huntingtin mRNA and protein levels in the *Hdac2* KO mice. In the RNA-seq data, *Hdac2* KO decreased total huntingtin mRNA levels in $Htt^{+/+}$ striata by 5.9% (nominal p=0.06; adjusted p=0.25) and in $Htt^{Q111/+}$ striata by 11% (nominal p<0.001; adjusted p<0.05) (*Figure 8—figure supplement 1*). However, western blot analyses on the contralateral striata from a subset of the mice used for RNA-seq did not provide evidence for an impact of *Hdac2* KO on steady state levels of wild-type or mutant huntingtin protein (*Figure 8—figure supplement 1C, D*). We also investigated the RNA-seq data for reads spanning exon 1 and intron 1, reflecting aberrantly spliced transcripts identified in a number of other HD models and predicted to encode exon1-containig N-terminal fragment of mutant huntingtin (*Sathasivam et al., 2013*). In support of previous data (*Sathasivam et al., 2013*), we identified exon1-intron1-spanning transcripts in $Htt^{Q111/+}$ striata. However, *Hdac2* KO had no impact on the number of these sequence reads (*Figure 8—figure supplement 1B*). It appears therefore that altered levels of mutant huntingtin or exon1 mutant huntingtin expressed from mis-spliced exon1-intron1 transcripts do not readily explain suppression of nuclear huntingtin pathology elicited by *Hdac2* KO. This is consistent with modification of these highly CAG length-dependent cellular phenotypes occurring as a consequence of reduced CAG repeat lengths in striatal MSNs. However, analyses of larger cohorts of mice as well as analyses of rare expansions not captured using GeneMapper would be needed to rigorously discern *Hdac2* and *Hdac3* genotype effects on nuclear huntingtin pathology phenotypes that are dependent on or independent of somatic CAG expansion.

## Discussion

Identification of genetic modifiers of disease phenotypes in HD mouse models provides an important route to dissecting underlying mechanisms of pathogenesis and ultimately for identifying therapeutic targets. Here, we have taken a cell type-specific genetic knockout approach to investigate the roles of *Hdac2* and *Hdac3* genes in as modifiers of molecular or cellular phenotypes elicited in MSNs by an expanded CAG repeat in heterozygous $Htt^{Q111}$ knock-in mice.

We restricted our analyses to previously described robust molecular and cellular changes in this model, specifically somatic instability, thought to drive the rate of disease onset in patients, transcriptional dysregulation and nuclear huntingtin pathology. We did not quantify neuronal cell loss and altered astrocytic or microglial cell density as these phenotypes are extremely subtle or not observed in this model. Similarly, we have not examined behavioral endpoints that are also subtle, as such studies would need significantly larger numbers of mice to be sufficiently powered.

Our results show that MSN-specific knockout of *Hdac2* reversed the transcriptional dysregulation of a subset of genes and resulted in a moderate reduction in somatic CAG expansion and nuclear huntingtin pathology. MSN-specific knockout of *Hdac3* resulted in a comparable reduction in somatic CAG expansion to the *Hdac2* knockout. The impact of genetic knock-out or reduction of several *Hdac* genes (*Hdac3*, *Hdac4*, *Hdac6*, *Hdac7*) has previously been investigated in the R6/2 exon 1 mouse model of HD and in *Hdh*Q150 knock-in mice with the goal of understanding which HDAC targets mediated the beneficial effects of broad-acting HDAC inhibitors (*Benn et al., 2009*;

*Bobrowska et al., 2011*; *Moumné et al., 2012*; *Mielcarek et al., 2013*). With the exception of *Hdac6*, the impact of the full knock-out could not be studied due to embryonic or early postnatal lethality. Of the genes tested, only *Hdac4* was found to modify mouse phenotypes, uncovering a novel role for this gene in modulating mutant huntingtin aggregation (*Mielcarek et al., 2013*). In R6/2 mice, the lack of any modifying effect of *Hdac3* may be due to the modest reduction in HDAC3 protein level (to 80% of wild-type overall, or to 60% of wild-type in the nucleus) in the *Hdac3*$^{+/-}$ mice (*Moumné et al., 2012*).

Strikingly, knockout of *Hdac2* in MSNs caused substantial transcriptional dysregulation, resulting in both up- and down-regulation of a large number of genes. These findings are consistent with data showing that HDAC1/2 can both positively and negatively regulate gene expression (*Kelly et al., 2018*). It is notable that the majority of mouse tissue-specific deletions of either *Hdac2* or *Hdac1* exhibited no discernible phenotype, presumably due to functional redundancy of the two highly related HDAC1 and HDAC2 proteins (*Kelly et al., 2018*). Our data reveal that, despite its up-regulation, HDAC1 is unable to compensate for the lack of HDAC2 in MSNs, providing evidence for a specific role of HDAC2 in MSNs and supporting distinct functions of HDAC1 and HDAC2 in adult neurons (*Guan et al., 2009*; *Kim et al., 2008*). The expression level of approximately 12% of genes dysregulated by the *Htt*$^{Q111}$ allele was normalized by the *Hdac2* KO to levels present in wild-type mice. However, this occurs against a much larger background of transcriptional dysregulation elicited by the *Hdac2* KO, with evidence for shared pathways and processes being impacted by both the *Htt*$^{Q111}$ and *Hdac2* knockout mutations. Indeed, a large proportion (~24%) of genes dysregulated by the *Htt*$^{Q111}$ allele was further dysregulated in the same direction by the *Hdac2* KO. It should be noted in interpreting these data that the majority of genes dysregulated by the *Htt*$^{Q111}$ allele did not meet a p value threshold corrected for multiple testing, although a large proportion of genes dysregulated by the *Hdac2* KO did. We therefore examined a core set of 29 genes significantly dysregulated (FDR adjusted p<0.05) by *Htt*$^{Q111}$ in both in the current study and in Langfelder et al., an extensive investigation of transcriptional dysregulation in HD knock-in mice *Langfelder et al., 2016*. Of these genes, only two met our criteria for rescue (*Ppp1r1b* and *Slc29a1*); for the other genes in this set, *Hdac2* KO either further altered the absolute gene expression level in the same direction as the *Htt*$^{Q111}$ allele or had no impact on expression. However, in these significantly dysregulated genes, we found that *Hdac2* KO lessened the relative impact of the *Htt*$^{Q111}$ allele, regardless of the absolute gene expression change elicited by the *Hdac2* KO. This genetic interaction indicates a functional relationship between these two mutations that may in part reflect a role of huntingtin in regulating the REST-containing transcriptional repressor complex that includes HDAC2 (*Buckley et al., 2010*).

Several studies have shown beneficial effects of HDAC inhibitors in HD mouse models, including broad-acting HDAC inhibitors SAHA, phenylbutyrate and parabinostat (LBH589), as well as compounds RGFP966 and 4b that show moderate selectivity towards HDAC3 (*Hockly et al., 2003*; *Gardian et al., 2005*; *Thomas et al., 2008*; *Mielcarek et al., 2011*; *Jia et al., 2012a*; *Jia et al., 2012b*; *Jia et al., 2016*; *Chopra et al., 2016*; *Suelves et al., 2017*; *Siebzehnrübl et al., 2018*). The impact on transcriptional dysregulation was investigated in a number of these studies, with most focusing on specific candidate genes. Overall, these studies revealed relatively modest reversal of transcriptional changes elicited by the mutant transgenes, though HDAC inhibitor 4b was found to normalize ~1/3 of genes altered in R6/2 mice harboring 300 CAG repeats (*Thomas et al., 2008*). Our data indicate that HDAC inhibitors with activity against HDAC2 would have the potential to reverse some of the early transcriptional changes elicited by an expanded CAG mutation. However, our results also indicate a complex relationship between the expanded CAG repeat mutation and *Hdac2*, with potential implications for therapies directed at HDAC2 inhibition.

We demonstrate here that *Hdac2* and *Hdac3* are enhancers of CAG expansion in MSNs. It is important to note that neither knockout of *Hdac2* nor *Hdac3* had a large impact on CAG expansion, in contrast to a number of MMR genes that are fundamental drivers of this process and whose knockouts eliminate expansion (*Wheeler et al., 2003*; *Dragileva et al., 2009*; *Pinto et al., 2013*). Our results are consistent with those of Suelves et al. in which treatment of *Htt*$^{Q111}$ mice with HDAC inhibitor RGFP966, resulted in a slight reduction in striatal CAG expansion (*Suelves et al., 2017*). HDAC3 and HDAC5 enhanced CAG expansion in a selectable human astrocyte cell-based assay, whereas HDAC9 suppressed expansions (*Debacker et al., 2012*; *Gannon et al., 2012*). A role for HDAC2 was not implicated in this system, based on differential effects of HDAC inhibitors with

selectivity towards HDACs 1 and 2 or HDAC3 (*Debacker et al., 2012*). Further studies are needed to understand the mechanism(s) by which *Hdac2* and *Hdac3* enhance CAG expansion in HD mice. There is evidence that CAG instability may depend on the chromatin environment in the vicinity of the repeat (*Libby et al., 2003*; *Libby et al., 2008*; *Nestor and Monckton, 2011*; *Neto et al., 2017*; *Goula et al., 2012*; *Dion and Wilson, 2009*), and both HDAC2 and HDAC3 play roles in regulating genome stability via effects on histone acetylation and recruitment or accessibility of DNA repair proteins (*Miller et al., 2010*; *Bhaskara et al., 2010*). Therefore, it is possible that *Hdac2* or *Hdac3* modify CAG instability by altering chromatin structure close to the CAG repeat tract. We did observe a small reduction in huntingtin mRNA in *Hdac2* knockout mice (*Figure 8—figure supplement 1*) suggesting a possible alteration in transcriptional dynamics at the *Htt* locus that may reflect an underlying change in chromatin structure. Our data argue, at least for the *Hdac2* KO, against altered transcription of genes encoding DNA repair proteins underlying the reduction in somatic expansion. However, it is possible that changes in acetylation of DNA repair proteins could lead to their altered activity (*Chatterjee et al., 2012*; *Choudhary et al., 2009*; *Zhang et al., 2014*; *Radhakrishnan et al., 2015*; *Piekna-Przybylska et al., 2016*; *Zhang et al., 2019*). This was proposed as a plausible mechanism by which *HDAC3* and *HDAC5* enhance CAG expansion in a selectable human astrocyte cell-based assay, in part based on epistasis experiments showing that *HDAC3* and *HDAC5* act in the same pathway as *MSH2* (*Debacker et al., 2012*; *Gannon et al., 2012*). It was recently suggested that HDAC3 might modify CAG instability by altering the subcellular localization of MSH3 (*Williams et al., 2020*). Understanding the interplay between HDACs and DNA repair proteins controlling CAG repeat dynamics may provide important insight into the regulation of these DNA repair proteins and potentially novel ways in which to intervene in the process of CAG expansion.

MSN-specific *Hdac2* knockout also reduced nuclear huntingtin intensity and inclusion phenotypes, consistent with decreased striatal CAG expansions and the known sensitivity of these phenotypes to repeat length and to modification of somatic instability (*Kovalenko et al., 2012*; *Dragileva et al., 2009*; *Pinto et al., 2013*; *Wheeler et al., 2002*). We did not observe a significant effect of *Hdac3* knockout on nuclear huntingtin intensity in the few mice analyzed and would likely need larger numbers of mice to observe a reduction in this phenotype that might be predicted by the lower striatal CAG repeat lengths in *Hdac3* KO mice. Most studies of pharmacological or genetic inhibition of class I HDACs in HD mouse models have reported a lack of impact on nuclear inclusions or aggregation (*Hockly et al., 2003*; *Gardian et al., 2005*; *Jia et al., 2016*; *Moumné et al., 2012*), although HDAC inhibitor 4b reduced aggregates in the N171-82Q model (*Jia et al., 2012b*). We suggest that *Hdac2* knockout modifies nuclear intensity and inclusion phenotypes in $Htt^{Q111}$ mice at least in part as a consequence of reduced CAG expansion in MSNs. Other mechanisms are also plausible, however, effects on mutant huntingtin levels themselves are not supported by our data. In contrast, the modest reduction in CAG expansion caused by the *Hdac2* knockout is unlikely to explain the reversal of transcriptional dysregulation elicited by the $Htt^{Q111}$ allele, in which gene expression levels that can be distinguished between wild-type and $Htt^{Q111/+}$ mice appear effectively normalized to wild-type in $Htt^{Q111/+}$ mice harboring the *Hdac2* knockout mutation.

In summary, we have tested whether *Hdac2* and *Hdac3* are modifiers of different aspects of the HD pathogenic process in MSNs that are both inherently prone to CAG repeat expansion and specifically vulnerable in the disease process. The results of this study indicate moderate effects of these genes on CAG expansion, which in humans drives the rate of disease onset, and for *Hdac2*, on downstream molecular and cellular phenotypes that may be important in cellular toxicity mechanisms.

## Materials and methods

### Mouse lines and breeding

$Htt^{Q111}$ knock-in mice used in this study were on a C57BL/6J background (*Lee et al., 2011*) and were maintained by breeding heterozygous males to C57BL/6J wild-type females from The Jackson Laboratory (Bar Harbor, ME). D9-Cre transgenic mice contain a genomic fragment, comprising about 2 kb of 5′ regulatory sequence, the endogenous ATG, and some of the introns and exons of the mouse *Ppp1r1b* gene encoding DARPP-32, driving the expression of Cre recombinase in MSNs after

5–6 weeks of age (*Bogush et al., 2005*). These mice were maintained on a C57BL/6J background. Mice harboring conditional 'floxed' *Hdac2* or *Hdac3* alleles contained *loxP* sites upstream of exon2/ downstream of exon 4 of the *Hdac2* gene (*Montgomery et al., 2007*) and upstream of exon 11/ downstream of exon 14 of *Hdac3* gene (*Montgomery et al., 2008*). Both floxed mouse lines were originally on a mixed C57BL/6/129SeEv background and subsequently maintained by breeding heterozygous males or females to C57BL/6J wild-type mice from The Jackson Laboratory.

Detailed breeding schemes are shown in *Figure 1—figure supplement 1*. Mice carrying conditional floxed *Hdac2* or *Hdac3* alleles were crossed with $Htt^{Q111}$ mice and, in a separate cross, with D9-Cre mice. The progeny from the two crosses were bred together to obtain $Htt^{+/+}$ and $Htt^{Q111/+}$ mice with or without the *Hdac2* (or *Hdac3*) floxed allele and/or Cre transgene. Mice harboring either no floxed *Hdac* alleles in the presence of Cre or one, two, or no floxed alleles in the absence of Cre were wild-type with respect to HDAC expression and are called *Hdac2* WT or *Hdac3* WT. Mice homozygous for the floxed *Hdac* alleles and expressing Cre are called *Hdac2* KO or *Hdac3* KO. In these mice the *Hdac2* or *Hdac3* alleles are deleted in MSNs and do not express HDAC2 or HDAC3 in MSNs. In a parallel cross, $Htt^{Q111/+}$ mice were crossed with D9-Cre mice to generate $Htt^{+/+}$ and $Htt^{Q111/+}$ mice with or without the Cre transgene, with the purpose of testing the effect of Cre expression alone on the phenotypes being measured. Note that mice were not fully backcrossed to C57BL/6J prior to the crosses with B6J.D9-Cre mice and B6J.$Htt^{Q111}$ mice. However, analyses of our RNA-seq data showed that all mice had a B6 haplotype for the *Mlh1* gene, where genetic variation between B6 and 129 strains is known to associate with CAG instability (*Pinto et al., 2013*).

Numbers of mice used in this study were gauged based on extensive prior knowledge of cellular and molecular phenotypes elicited by the $Htt^{Q111}$ allele and their genetic modification (*Kovalenko et al., 2012*; *Pinto et al., 2013*; *Langfelder et al., 2016*), and incorporated controlling for inherited CAG repeat length (*Source data 1*). In all the mouse experiments, 'N'=number of mice (biological replicates).

## Genotyping

Genomic DNA was extracted from tail or ear biopsies taken at weaning for genotyping, or from adult striatum for the CAG repeat instability assay (Qiagen DNeasy blood and tissue kit). Genotypes were further confirmed in striatal DNA. CAG sizing was carried out as previously described, using human *HTT* CAG-specific primers (*Dragileva et al., 2009*). PCR products were resolved using the AB13730xl automated DNA analyzer (Applied Biosystems). GeneMapper v3.7 with GeneScan 500-LIZ as internal size standard was used to assign repeat size, defined as the highest peak in the GeneMapper trace. All runs included the same control DNAs of known *HTT* CAG repeat size. We incorporated a novel PCR genotyping assay that distinguished knock-in and mouse wild-type alleles using primers flanking the 5' *loxP* site remaining after excision of the floxed neomycin resistance cassette used in the original gene targeting (*White et al., 1997*; *Wheeler et al., 1999*). The assay included 0.5 µM forward primer (5′ TCTGATACAGCCCATGCTGA), 0.5 µM reverse primer (5′ CTGAGTTCAA TCCCTGCATTT) and 2x Phusion High-Fidelity PCR Master Mix with HF Buffer (New England Biolabs). Cycling conditions were 98°C for 30 s; 30 cycles of 98°C for 5 s, 62°C for 15 s, 72°C for 20 s; 72°C for 10 min, generating a 129 bp product from the wild-type allele and a 240 bp product from the $Htt^{Q111}$ knock-in allele.

D9-Cre mice were genotyped as described (*Bogush et al., 2005*). For *Hdac2* genotyping, wild-type and floxed alleles were detected by a triplex PCR assay. The forward primer upstream of 5′ *loxP* site (HDAC2WT-F: 5′ GGCCAAGCATATTCAAACCACC) and the reverse primer between the 5′ *loxP* site and exon 2 (HDAC2-REV: 5′ GTCAGCTAGTAGTGCTTCTTGG) generated a 226 bp product for the wild type *Hdac2* allele, while the forward primer within the 5′ *loxP* site (HDAC2MUT-F; 5′ G TCCCTCGACCTGCAGGAATTC) and the same reverse primer gave a 146 bp PCR product for the floxed allele. The PCR assay contained 0.2 µM each of HDAC2WT-F and HDAC2MUT-F primers, 0.4 µM of HDAC2-REV primer, 0.025 units GoTaq DNA polymerase (Promega), 5x GoTaq buffer (Promega), additional $MgCl_2$ (5 mM), and dNTP mixture (0.05 mM each). Cycling conditions were 94°C for 2 min; 30 cycles of 94°C for 15 s, 60°C for 30 s, 72°C for 40 s; 72°C for 5 min. To detect specifically the deleted *Hdac2* allele, a different forward primer upstream of 5′ *loxP* site (5′ GTGGGAAGCA TGGCAGCATGC) and reverse primer downstream of the 3′ *loxP* site, between exons 4 and 5 (5′ GCCTTCTAAGAACCCCAGGGAAC) were used, resulting in a 550 bp product upon Cre-mediated excision. PCR conditions were 0.2 µM of each primer, 0.025 units GoTaq DNA polymerase

(Promega), 5x GoTaq buffer (Promega), additional $MgCl_2$ (5 mM), and dNTP mixture (0.05 mM each). Cycling conditions were 94℃ for 2 min; 30 cycles of 94℃ for 15 s, 68℃ for 30 s, 72℃ for 40 s; 72℃ for 5 min. For *Hdac3* genotyping, a triplex assay was used to detect all three alleles, with one forward/reverse pair of primers up- and downstream from the 5′ *loxP* site (HDAC3-5′-loxP-F: 5′ GC TTGGTAGCCAGCCAGCTTAG, and HDAC3-5′-loxP-R: 5′ CATGTGACCCCAGACATGACTGG), and the third (reverse) primer downstream of the 3′ *loxP* site (HDAC3-3′-loxP-R: 5′ CAGTCCATGCCTA TAATCCCAGC). This assay produced a 350 bp product for the wild-type *Hdac3* allele, a 450 bp product for the 'floxed' allele (both products were from HDAC3-5′-loxP-F and HDAC3-5′-loxP-R primers), and a 650 bp product for the deleted *Hdac3* allele after Cre-mediated excision (from HDAC3-5′-loxP-F and HDAC3-3′-loxP-R primers). The PCR assay contained 0.2 µM each of the three primers, 0.025 units GoTaq DNA polymerase (Promega), 5x GoTaq buffer (Promega), additional $MgCl_2$ (5 mM), and dNTP mixture (0.05 mM each). Cycling conditions were 94℃ for 2 min; 30 cycles of 94℃ for 15 s, 60℃ for 30 s, 72℃ for 40 s; 72℃ for 5 min.

## Measurement of CAG expansion

Expansion indices in striatum were quantified from GeneMapper traces of PCR-amplified *HTT* CAG repeat, as described previously (*Lee et al., 2010*), taking into account only expansion peaks to the right of the highest peak (main allele) and without filtering peaks based on relative peak height. Although there is intrinsic PCR bias in analyses of instability from 'bulk DNA', this nevertheless allows quantification of subtle differences in repeat instability, and metrics derived from the GeneMapper traces correlate well with repeat length distributions obtained from analyses of single molecules (*Lee et al., 2010*). Briefly, normalized peak heights were calculated by dividing each expansion peak height by the sum of the heights of the main allele plus all expansion peaks, the change in CAG length of each expansion peak from the main allele was determined, each normalized peak height was multiplied by the CAG change from the main allele, and these values were summed to generate an expansion index that represents the mean positive CAG repeat length change in the population of cells being analyzed.

## Immunohistochemistry

Primary antibodies used were: mouse monoclonal anti-huntingtin (mAb5374, Millipore), rabbit polyclonal anti-histone H3 (ab1791, Abcam), rabbit polyclonal anti-HDAC2 (ab7029, Abcam), rabbit monoclonal anti-HDAC3 (clone Y415; formerly available from Millipore), mouse monoclonal anti-DARPP-32 (D32-6a; a kind gift from Dr. Angus Nairn) and rabbit polyclonal anti-DARPP-32 (ab40801, Abcam).

Immunostaining was performed on 8 µm coronal sections of periodate-lysine-paraformaldehyde (PLP)-perfused and -post-fixed, paraffin-embedded brains. Perfusion and tissue processing/embedding methods have been described previously (*Wheeler et al., 2000*). For most mouse cohorts, one hemisphere of the brain was embedded for sectioning and the other one was used for dissection of striatum for analysis of somatic instability. Sections chosen for immunostaining were aligned with respect to their anterior/posterior location in the brain. The sections were deparaffinized, rehydrated and subjected to heat-mediated epitope retrieval (Na-citrate buffer pH 6.0) followed by quenching of endogenous peroxidase with 0.3% $H_2O_2$/methanol for 30 min at room temperature (RT), and blocked in 3% normal horse serum (NHS) in Tris-buffered saline (TBS) for 1 hr at room temperature. Immunostaining for diffuse nuclear huntingtin, huntingtin inclusions and histone H3 was described in detail (*Kovalenko et al., 2012*). For double staining with mouse anti-DARPP-32/rabbit anti-HDAC3 antibodies sections were incubated with both primary antibodies (D32-6a at 1:200 and Y415 at 1:2000 in 1% NHS/TBS) overnight at 4℃, followed by sequential amplification of HDAC3 and DARPP-32 signal. First, HDAC3 signal was amplified using the TSA Biotin System (Perkin Elmer) according to manufacturer's instructions and quenching of HRP activity (0.1% sodium azide/0.3% $H_2O_2$ in TBS for 30 min at RT) was performed before the addition of streptavidin-Alexa Fluor 555. Then, DARPP-32 signal was amplified using TSA kit #2 (Invitrogen/Thermo Fisher Scientific) by incubating with goat anti mouse-HRP at 1:100 for 1 hr at RT and then with Tyramide-Alexa Fluor-488 at 1:100 in the amplification diluent (all provided with the TSA kit #2), for 25 min at RT. For double staining with mouse anti-DARPP-32/rabbit anti-HDAC2 antibodies sections were incubated with both primary antibodies (D32-6a at 1:200 and ab7029 at 1:1000 in 1% NHS/TBS) overnight at 4℃.

DARPP-32 signal was amplified as described above, and HDAC2 signal was detected with donkey anti-rabbit Alexa Fluor 555-conjugated secondary antibody (Invitrogen/Thermo Fisher Scientific, 1:1000) added as the last step after DARPP-32 signal amplification. For staining with rabbit anti-DARPP-32 antibody sections were incubated with ab40801 at 1:750 at 4˚C overnight, followed by donkey anti-rabbit Alexa Fluor 488 secondary antibody.

All slides were mounted in ProLong Gold antifade reagent (Invitrogen/Thermo Fisher Scientific). Fluorescent microscopy was performed with a Zeiss Axioskop two microscope equipped with Axio-CamMRm camera and AxioVision 4.6 image acquisition software, using Plan Apochromat 20x/0.8 M27 or Plan Neofluar 40x/0.75 Ph2 objectives. Images that were to be quantified and compared were taken with the same exposure times.

## Quantification of immunohistochemical data

Quantification was performed as described previously (*Kovalenko et al., 2012*), using CellProfiler 3.0.0 image analysis software (*Carpenter et al., 2006*). Briefly, diffuse nuclear huntingtin immunostaining was quantified in four 40x striatal images per mouse (one medial, located next to the lateral ventricle, and one lateral, located at the end of external capsule; from two consecutive sections). Total (integrated) intensity of mAb5374 staining was measured in all mAb5374-positive nuclei in each image and normalized by the total number of nuclei (as determined by the number of all histone H3-positive nuclei) in the same image. The resulting value, representing the mean intensity of mAb5374 staining per nucleus, was averaged from four images for each mouse. The number of nuclear huntingtin inclusions (total per image) was quantified, blind to genotype, with CellProfiler 3.0.0 in two 20x images per mouse (taken mid-striatum from two consecutive coronal sections), normalized by the total number of histone-H3-positive nuclei in each image (taken as 100%), and resulting percentages were averaged over these two images for each mouse. Number of DARPP-32-positive cells and total integrated intensity of DARPP-32 staining in DARPP-32-positive cells were also quantified with CellProfiler 3.0.0 in four 40x images per mouse, taken in the same striatal locations as for diffuse nuclear huntingtin immunostaining quantification. Total integrated intensity was normalized by the number of DARPP-32-positive cells in each image to obtain mean integrated intensity values. Both mean integrated intensity of DARPP-32 staining and the number of DARPP32-positive cells were averaged from four images for each mouse.

## Western blot analysis

Striatal extracts were prepared as described previously (*Kovalenko et al., 2012*). For detection of huntingtin protein, 90 µg of striatal extracts were resolved by SDS-PAGE in Novex 3–8% Tris-acetate gels (Invitrogen/Thermo Fisher Scientific) and transferred to 0.45 µm nitrocellulose membrane (Bio-Rad) in Transfer Buffer (Boston BioProducts) at 100 V for 72 min. After blocking with 5% non-fat milk, HTT protein was detected using mAb2166 antibody (Millipore) at 1:1000 in 5% non-fat milk/TBS/0.05% Tween-20 and visualized with ECL kit (ThermoScientific).

For HDAC2 and HDAC1, 40 µg of striatal extracts were resolved on 12% Bis-Tris gels (Invitrogen/Thermo Fisher Scientific) and transferred in Transfer Buffer with 10% methanol, 0.025% SDS at 110 V for 75 min. The membranes were blocked with 5% non-fat milk in TBS/0.05% Tween-20 and incubated with anti-HDAC2 H3159 antibody (Sigma) at 1:10000 or anti-HDAC1 ab31263 (Abcam) at 1:2000 in 1% non-fat milk/TBS/0.05% Tween-20 and developed with ECL kit.

For detection of acetylated histones H3AcK9 and H4AcK12, 35 µg of striatal extracts were resolved on 12% Bis-Tris gels (Invitrogen/Thermo Fisher Scientific) and transferred in Transfer Buffer with 10% methanol, 0.025% SDS at 150 V for 75 min. The membranes were blocked as above and incubated with anti-H3AcK9 ab10812 (Abcam) at 1:500 or anti-H4AcK12 #04–119 (Millipore) at 1:500 in 1% non-fat milk/TBS/0.01% Tween-20 and developed with ECL kit.

Total histone H3 was detected after stripping and re-probing the same membrane with ab1791 (Abcam) at 1:2000 in 1% non-fat milk/TBS/0.05% Tween-20.

For detection of MSH3, SDS-PAGE and transfer was as for HDAC1 and HDAC2 (above). The membranes were blocked as above and incubated with anti-MSH3 mouse monoclonal antibody 2F11 -a kind gift from Glen Morris and Ian Holt (*Holt et al., 2011*)- at 1:300 in 5% non-fat milk/TBS/0.05% Tween-20.

For quantification of Western blot data the density of protein bands of interest was measured with Quantity One software (Bio-Rad) with local background subtraction and normalized by the total amount of protein per corresponding lane as measured by the density of the entire lane stained by Novex reversible protein stain (Invitrogen/Thermo Fisher Scientific). Normalized values (one per mouse) were averaged over each genotype group.

## Selectable cell-based assay for CAG repeat contractions

The selectable assay for repeat contractions, based on insertion of a CAG repeat into an intron of an *HPRT* minigene, was performed in FLAH25 cells as described (*Hubert et al., 2011*). Cell maintenance, inhibitor treatment regimen, and selection for resistant colonies in HAT medium (0.1 mM hypoxanthine, 0.4 µM aminopterin, and 16 µM thymine) were as described (*Hubert et al., 2011*).~$4.5 \times 10^6$ cells were plated and the number of HPRT$^+$ colonies was adjusted by cell survival ([number of positive colonies] / [number of plated cells multiplied by plating efficiency]). The effect of inhibitors on CAG repeat contractions was expressed as fold change of the adjusted number of positive colonies over DMSO.

## RNA-seq library preparation and sequencing

RNA was isolated from striatal tissues of 36 5-month mice of four genotype groups: $Htt^{+/+}$ $Hdac2$ WT (N = 9: six males, three females), $Htt^{Q111/+}$ $Hdac2$ WT (N = 7: four males, three females), $Htt^{+/+}$ $Hdac2$ KO (N = 9: five males, four females) and $Htt^{Q111/+}$ $Hdac2$ KO (N = 11: eight males, three females) lines using TRIzol reagent. Briefly, pelleted cells were resuspended in TRIzol reagent and then extracted with chloroform, followed by isopropanol precipitation of RNA from the aqueous phase and three 70% ethanol washes. RNA pellets were solubilized in 30–50 µl of RNase-free water (Ambion, AM9937). RNA quality was assessed using the Agilent Bioanalyzer TapeStation 2200 (Agilent Technologies, Santa Clara, CA). In total, 36 RNA-seq libraries were prepared using the Illumina TruSeq Stranded mRNA Sample Prep Kit. Each library in this study included 1 ul of a 1:10 dilution of ERCC RNA Control Spike-Ins (Ambion) that were added from one of two mixes, each containing the same 92 synthetic RNA standards of known concentration and sequence. These synthetic RNAs cover a $10^6$ range of concentration, as well as varying in length and GC content to allow for validation of dose response and the fidelity of the procedure in downstream analyses (*Jiang et al., 2011*). Libraries were multiplexed, pooled, and sequenced on multiple lanes of an Illumina HiSeq2000, generating median 40.7M paired-end reads per library of 75 bp.

## RNA-seq data processing and analysis

Quality checking of sequence reads was assessed using fastQC (v. 0.10.1) (http://www.bioinformatics.babraham.ac.uk/projects/fastqc), blind to genotype. Sequence reads were aligned to the mouse reference genome (GRCm38, Ensembl build v. 75) using GSNAP (version 2015-06-23) with parameters '-a off -N 1 –pairexpect=450 -B 3 –query-unk-mismatch=1' (*Wu and Nacu, 2010*). Gene level counts were tabulated using BedTools's multibamcov algorithm on unique alignments for each library at all Ensembl genes (GRCm38 v.75). Based on quality checking of alignments assessed by custom scripts utilizing PicardTools (https://broadinstitute.github.io/picard/), RNASeQC (*DeLuca et al., 2012*), RSeQC (*Wang et al., 2012*) and SamTools (*Li et al., 2009*), one male sample with genotype $Htt^{+/+}$ $Hdac2$ WT was identified as an outlier and thus excluded from further analysis as this sample showed high intergenic (25%), intronic (23%), duplication rates (57%) and low exonic rates (53%), which indicated DNA contamination in this sample. Analysis of ERCC spike-ins as described (*Blumenthal et al., 2014*) estimated the expression threshold for detection to be at least six uniquely mapped reads. Genes smaller than 250 nt in length and tRNA and rRNA genes were removed from further analyses. Principal component analysis (PCA) of the remaining 35 samples was performed using a Bioconductor package, DESeq2 (v. 1.18.1) (*Love et al., 2014*) relying on regularized log-transformed counts after keeping genes expressed in all the samples from at least one genotype group with at least six counts. Differentially expressed genes (DEGs) in five pair-wise comparisons $Htt^{+/+}$ $Hdac2$ KO vs. $Htt^{+/+}$ $Hdac2$ WT [contrast 1], $Htt^{Q111/+}$ $Hdac2$ KO vs. $Htt^{Q111/+}$ $Hdac2$ WT [contrast 2], $Htt^{Q111/+}$ $Hdac2$ WT vs. $Htt^{+/+}$ $Hdac2$ WT [contrast 3], $Htt^{Q111/+}$ $Hdac2$ KO vs. $Htt^{+/+}$ $Hdac2$ KO [contrast 4] and $Htt^{Q111/+}$ $Hdac2$ KO vs. $Htt^{+/+}$ $Hdac2$ WT [contrast 5], were identified by edgeR's quasi-likelihood F test (v. 3.12) (*Robinson et al., 2010*) with surrogate variables (SVs)

accounting for batch effects and unknown factors in data, which were determined by Bioconductor Package SVA (v. 3.18) (*Leek et al., 2012*; *Leek, 2014*). To this end, SVA was applied to all 35 RNA-seq samples separately for each comparison of interest where the full model was mod = ~ 0 + genotype_group, while the null model was mod0 = ~ 1, analyzing genes that passed the expression detection threshold ($\geq$ 6 counts), in at least one genotype group samples in a given comparison. As the number of analyzed genes differed from comparison to comparison, SVA identified a distinct set of SVs for each comparison, although they were highly similar between comparisons (average correlation = 0.99). DEGs were defined at two statistical significance levels: less stringent, using a nominal $p < 0.05$ cut-off, and more stringent, using a false discovery rate (FDR) cut-off of $< 0.05$, calculated based on the Benjamini-Hochberg procedure (*Benjamini and Hochberg, 1995*). KEGG pathway and gene ontology (GO) enrichments were performed using DAVID (v. 6.8) for DEGs at both $p < 0.05$ and FDR $< 0.05$ levels (*Huang et al., 2009*). In doing so, lists of DEGs in either category of direction of dysregulation (upregulated, downregulated and both) for a given comparison were used as a query list, while a list of all the analyzed genes was treated as a background list. Differentially regulated pathways were also defined at less stringent (p<0.05) and more stringent (FDR<0.05) significance levels. We compared our differential expression results from the $Htt^{Q111/+}$ $Hdac2$ WT vs $Htt^{+/+}$ $Hdac2$ WT comparison (contrast 3) with those from a comparison of 6 month $Htt^{Q111/+}$ (4 female, 4 male) vs $Htt^{+/+}$ (4 female, 4 male) striata using the RNA-Seq dataset described in *Langfelder et al., 2016*. To this end, we reanalyzed the Langfelder RNA-Seq data employing the same EdgeR + SVA pipeline as in the present study (as above). 16,395 genes that passed our expression threshold ($\geq$ 6 counts) in either all mutant or all control samples were analyzed. We compared DEGs identified in both studies at both $p < 0.05$ and FDR $< 0.05$ thresholds as above. To assess the statistical significance of overlaps between DEGs or enriched pathways between pairs of contrasts (*Figure 3—figure supplement 1*) we performed one-tailed Fisher's Exact tests, commonly used to assess the statistical significance of overlap of two sets or enrichment of one set in other (overrepresentation). To identify genes where $Hdac2$ knockout is either protective or exacerbates the effect of the $Htt^{Q111}$ allele, we first compared DEGs (p < 0.05) in contrast 3 ($Htt^{Q111/+}$ vs. $Htt^{+/+}$ in $Hdac2$ WT background) with those in contrast 2 ($Hdac2$ KO vs. WT in $Htt^{Q111/+}$ background). Next, we performed 2-tailed unpaired t-tests on those shared DEGs to identify genes that were not significantly differentially expressed (p > 0.05) between $Htt^{+/+}$ $Hdac2$ WT and $Htt^{Q111/+}$ $Hdac2$ KO. To assess the statistical significance of observing reduced relative expression (fold-change of $Htt^{Q111}$ versus $Htt^{+/+}$) of the 29 commonly dysregulated genes shared between thus study and Langfelder et al. in the Hdac2 KO background compared to the Hdac2 WT background we performed a permutation test. To this end, we randomly selected 29 genes 100,000 times from 15,508 genes analyzed in both this study and Langfelder et al. In each iteration, we compared the absolute value of log2 fold-changes of genes between $Htt^{Q111/+}$ and $Htt^{+/+}$ genotypes in $Hdac2$ KO and $Hdac2$ WT backgrounds and assessed how many times the randomly selected 29 genes showed the observed effect, that is $abs\left(log2FC^i_{Hdac2\ KO}\right) < abs\left(log2FC^i_{Hdac2\ WT}\right)$, where $abs\left(log2FC^i_{Hdac2\ KO}\right)$ is absolute value of log2 fold-change of $i^{th}$ gene between $Htt^{Q111/+}$ and $Htt^{+/+}$ genotypes in $Hdac2$ KO background. All analyses were run on R platforms (versions 3.2.2, 3.3.2, 3.4.3, 3.6).

## Statistical analyses

We performed statistical analyses to compare quantitative instability and mAb5374 phenotypes between genotype groups. As we were not testing the effect of age on these phenotypes, and as our datasets were well controlled for inherited CAG length, we performed two-tailed unpaired t-tests to compare mean values between two genotype groups at any one age. Statistical analyses relevant to the RNA-seq data analyses are described in the section above.

## Acknowledgements

We are very grateful to Dr. Michelle Ehrlich for providing the DARPP-32 Cre transgenic mice, and to Dr. Eric Olson for providing the $Hdac2$ and $Hdac3$ floxed mice. We would like to thank CHDI for providing the 'allelic series' RNA-seq data described in *Langfelder et al., 2016*.

## Additional information

### Competing interests

Daniel M Fass: DMF is a member of the scientific advisory board of Psy Therapeutics. Stephen J Haggarty: SJH is a member of the scientific advisory board of Psy Therapeutics, Frequency Therapeutics and Souvien Therapeutics, and former member of the scientific advisory board of Rodin Therapeutics that is focused on HDAC2 inhibitors, none of whom were involved in the present study. SJH has also received speaking or consulting fees from Amgen, AstraZeneca, Biogen, Merck, Regenacy Pharmaceuticals, as well as sponsored research or gift funding from AstraZeneca, JW Pharmaceuticals, and Vesigen unrelated to the content of this manuscript. His financial interests were reviewed and are managed by Massachusetts General Hospital and Partners HealthCare in accordance with their conflict of interest policies. Vanessa C Wheeler: VCW is a scientific advisory board member of Triplet Therapeutics, a company developing new therapeutic approaches to address triplet repeat disorders such Huntington's disease and Myotonic Dystrophy and of LoQus23 Therapeutics. Her financial interests in Triplet Therapeutics, were reviewed and are managed by Massachusetts General Hospital and Partners HealthCare in accordance with their conflict of interest policies. The other authors declare that no competing interests exist.

### Funding

| Funder | Grant reference number | Author |
|---|---|---|
| Huntington Society of Canada | New Pathways Research Grant | Vanessa C Wheeler |
| National Institutes of Health | NS049206 | Vanessa C Wheeler |
| Huntington's Disease Society of America | Berman Topper Career Development Award | Ricardo Mouro Pinto |
| National Institutes of Health | GM38219 | John H Wilson |
| National Institutes of Health | EY11731 | John H Wilson |
| National Institutes of Health | 1F3HG004918 | Leroy Hubert |

The funders had no role in study design, data collection and interpretation, or the decision to submit the work for publication.

### Author contributions

Marina Kovalenko, Conceptualization, Data curation, Formal analysis, Supervision, Validation, Investigation, Visualization, Methodology, Writing - original draft, Writing - review and editing; Serkan Erdin, Conceptualization, Resources, Data curation, Formal analysis, Validation, Investigation, Visualization, Methodology, Writing - original draft; Marissa A Andrew, Leroy Hubert, Formal analysis, Investigation, Writing - review and editing; Jason St Claire, Melissa Shaughnessey, Investigation, Writing - review and editing; João Luís Neto, Investigation, Methodology, Writing - review and editing; Alexei Stortchevoi, Investigation; Daniel M Fass, Resources, Writing - review and editing; Ricardo Mouro Pinto, Formal analysis, Funding acquisition, Methodology, Writing - review and editing; Stephen J Haggarty, Conceptualization, Resources, Supervision, Writing - review and editing; John H Wilson, Michael E Talkowski, Conceptualization, Supervision, Writing - review and editing; Vanessa C Wheeler, Conceptualization, Data curation, Formal analysis, Supervision, Funding acquisition, Validation, Visualization, Methodology, Writing - original draft, Project administration, Writing - review and editing

### Author ORCIDs

Serkan Erdin  http://orcid.org/0000-0001-6587-2625
João Luís Neto  http://orcid.org/0000-0003-0863-158X
Daniel M Fass  https://orcid.org/0000-0003-0018-8093
Stephen J Haggarty  https://orcid.org/0000-0002-7872-168X
Vanessa C Wheeler  https://orcid.org/0000-0003-2619-589X

### Ethics

Animal experimentation: This study was carried out in accordance with the recommendations in the Guide for the Care and Use of Laboratory Animals of the National Institutes of Health under an approved protocol (2009N000216) of the Massachusetts General Hospital Subcommittee on Research Animal Care.

### Decision letter and Author response

Decision letter https://doi.org/10.7554/eLife.55911.sa1
Author response https://doi.org/10.7554/eLife.55911.sa2

## Additional files

### Supplementary files

- Source code 1. RNA-seq codes and data files.
- Source data 1. CAG lengths of mouse cohorts.
- Source data 2. Differentially expressed genes.
- Source data 3. Pathway analyses.
- Source data 4. Rescue analysis.
- Transparent reporting form

### Data availability

RNA-Seq data is deposited in GEO, under the accession number GSE148440.

The following dataset was generated:

| Author(s) | Year | Dataset title | Dataset URL | Database and Identifier |
|---|---|---|---|---|
| Kovalenko M, Erdin S, Andrew MA, Claire J, Shaughnessey M, Hubert L, Neto JL, Stortchevoi A, Fass DM, Pinto RM, Haggarty SJ, Wilson JH, Talkowski ME, Wheeler VC | 2020 | Histone deacetylase knockouts modify transcription, CAG instability and nuclear pathology in Huntington disease mice | https://www.ncbi.nlm.nih.gov/geo/query/acc.cgi?acc=GSE148440 | NCBI Gene Expression Omnibus, GSE148440 |

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
