## [Decision Letter]

**Acceptance summary:**

This manuscript provides novel data that histone deacetytlase 2, HDAC2, impacts transcriptional dysregulation found in striatal medium-spiny neurons affected with Huntington's disease. Loss of HDAC2 function is shown to normalize a subset of transcriptionally dysregulated genes associated with HD.

**Decision letter after peer review:**

Thank you for submitting your manuscript "Molecular and cellular consequences of genetic knockout of *Hdac2* and *Hdac3* in striatal medium spiny neurons of Huntington's disease knockin mice" for consideration by *eLife*. Your article has been reviewed by three peer reviewers, one of whom is a member of our Board of Reviewing Editors, and the evaluation has been overseen by K VijayRaghavan as the Senior Editor. The reviewers and the Reviewing Editor drafted this decision letter to help you prepare a revised submission.

This manuscript provides novel data that histone deacetytlase 2, HDAC2, impacts transcriptional dysregulation found in striatal medium-spiny neurons affected with Huntington's disease. Loss of HDAC2 function is shown to normalize a subset of transcriptionally dysregulated genes associated with HD.

Essential Revisions:

1) The submitted version is directed at the assessing the ability of *Hdac2* and *Hdac3* deletion to modify both cellular and molecular phenotypes associated with HD. As the authors note, the data hint the *Hdac2* deletion modifies HD phenotypes via mechanisms unrelated to CAG expansion. The inability of the data to firmly interpret the relationship between instability and striatal phenotypes detracts from the impact of the current version. All reviewers found the most compelling data to be the impact of HADC loss-of-function on alterations in gene expression associated with HD. Thus, it is critical that the manuscript be revised to focus on HDAC and medium-spiny neuron transcriptional regulation. Perhaps the effect of HADCs on somatic instability would best be presented not as the lead but as a possible mechanism by which dysregulated genes associated with HD are normalized. Likewise, the title should be reconfigured to reflect focus on HDACs and MSN transcriptional dysregulation in HD.

2) There is a lack of mechanism provided for the observed molecular changes. The Manuscript proposes a potential mechanism for how HDACs may be contributing to stability by modifying chromatin but do not provide evidence to support this. It is alternatively suggested that KO could affect acetylation of repair proteins. The data presented does not really support this proposed mechanism as there is no change in expression of DDR genes at transcript level and one (MSH3) at the protein level. It would have been helpful to assess levels of unmodified or acetylated MSH2 or MLH1 in the manuscript as these are known modifiers of somatic instability and were suggested in the Discussion. There is a description of an antibody for detecting MSH2 on western blots in the Materials and methods, however this data does not appear in the manuscript nor in the supplement. This could support data on potential mechanisms. Therefore it is recommended to revise the manuscript overall to prioritize the transcriptional normalization that occurs from KD and focus on those molecular insights with reduced focus on the somatic repeat instability piece. At present there is not a direct mechanism on somatic repeat instability and *Hdac2* KD and data provided for this possibility in the manuscript would suggest this may not be the case (no changes in DDR genes or levels of MSH3 protein).

3) The methodology regarding how the mice colonies were made and maintained is confusing. Are these mice heterozygous or homozygous for Q111? Perhaps a supplemental breeding scheme diagram? This is vital for data interpretation. Also, please include the Q111 genotype in the main text as well as in the Materials and methods.

4) Presentation of RNAseq data is difficult to follow and not fully described. For instance, it is not clear from the text whether genes that are consistently altered in HD are restored in expression and information on statistical enrichment. Similarly, the comparison with Langfelder et al., data is unclear. Suggest to use graphical visuals representing gene changes and observed overlaps instead of tables or guide readers' eye to specific rows and columns being discussed when referencing figures. Further, suggest adding additional figure references in the fourth paragraph of the subsection “Impact of MSN-specific *Hdac2* knockout on the transcriptome” for RNAseq comparisons to make it more clear so to which datasets text is referring to. Finally, conclusions relating to analysis from RNAseq does not appear to be consistent with the data. While the manuscript states that none of the rescued genes met the adjusted p-value cut-off (subsection “A subset of genes dysregulated in *Htt*^Q111^ striata is reversed by knockout of *Hdac2* in MSNs”), the author's interpretation of results suggests a rescue effect (subsection “A subset of genes dysregulated in *Htt*^Q111^ striata is reversed by knockout of *Hdac2* in MSNs”, last paragraph). Due to the lack of significance, would recommend re-phrasing to a suggested rescue rather than a definite one.

5) While the Discussion addresses some of the previous studies investigating HDACs and HDAC inhibitors in HD, this is by no means comprehensive. Studies by Hoshino et al., 2003, and Qunti et al., 2010, should be included discussing HDAC expression in HD human and mouse tissue, respectively. Additionally, work by Jia et al., 2016, Chopra et al., 2016, and Siebzehnrubl et al., 2018, discuss the effects of various HDAC inhibitors on models of HD. Please include these in your Discussion section and also give a brief introduction to this previous work in the Introduction.

---

## [Author Response]

Essential Revisions:1) The submitted version is directed at the assessing the ability of Hdac2 and Hdac3 deletion to modify both cellular and molecular phenotypes associated with HD. As the authors note, the data hint the Hdac2 deletion modifies HD phenotypes via mechanisms unrelated to CAG expansion. The inability of the data to firmly interpret the relationship between instability and striatal phenotypes detracts from the impact of the current version. All reviewers found the most compelling data to be the impact of HADC loss-of-function on alterations in gene expression associated with HD. Thus, it is critical that the manuscript be revised to focus on HDAC and medium-spiny neuron transcriptional regulation. Perhaps the effect of HADCs on somatic instability would best be presented not as the lead but as a possible mechanism by which dysregulated genes associated with HD are normalized. Likewise, the title should be reconfigured to reflect focus on HDACs and MSN transcriptional dysregulation in HD.

This was very helpful, and we agree that the manuscript needed to be refocused in a different way. To do this, we have reframed the Introduction to highlight a “two-step” mechanism of HD pathogenesis, supported by recent GWAS studies, invoking both CAG expansion and “cellular toxicity” as two critical components in HD. These two components are separable mechanistically, and therefore can be influenced by different modifiers, or by the same modifier by different mechanisms. However, modifiers of CAG expansion may alter downstream processes that relate to cellular toxicity if there is sufficient sensitivity to detect the effects of somatically expanded repeats (as is the case for nuclear huntingtin pathology phenotypes). This framework therefore does not set up the expectation that effects on transcriptional dysregulation should necessarily be influenced by somatic expansion, but rather that modifier effects can be independent. Our interpretation that these are likely to be independent is in the Discussion.

At the recommendation of the reviewers we have presented the *Hdac2* KO RNA-seq data first, in order to emphasize these data. This also allows us to better refer back to these data in subsequent sections where we specifically examine expression levels of specific genes, as candidate hypotheses in relation to modifier effects on CAG instability and nuclear huntingtin pathology.

We have modified the title to specifically include effect on transcription, but also to include other endpoints being measured.

2) There is a lack of mechanism provided for the observed molecular changes. The Manuscript proposes a potential mechanism for how HDACs may be contributing to stability by modifying chromatin but do not provide evidence to support this. It is alternatively suggested that KO could affect acetylation of repair proteins. The data presented does not really support this proposed mechanism as there is no change in expression of DDR genes at transcript level and one (MSH3) at the protein level. It would have been helpful to assess levels of unmodified or acetylated MSH2 or MLH1 in the manuscript as these are known modifiers of somatic instability and were suggested in the Discussion. There is a description of an antibody for detecting MSH2 on western blots in the Materials and methods, however this data does not appear in the manuscript nor in the supplement. This could support data on potential mechanisms. Therefore it is recommended to revise the manuscript overall to prioritize the transcriptional normalization that occurs from KD and focus on those molecular insights with reduced focus on the somatic repeat instability piece. At present there is not a direct mechanism on somatic repeat instability and Hdac2 KD and data provided for this possibility in the manuscript would suggest this may not be the case (no changes in DDR genes or levels of MSH3 protein).

We appreciate that this manuscript does not provide mechanistic insight into the instability modification, and meant only to propose certain hypotheses, rather than specifically favor one over the other. However, as we are able to test mRNA levels of DNA repair genes, we are reasonably able to conclude that *Hdac2* KO does not alter their expression. We followed up on one (MSH3) with a western blot, based on slightly reduced mRNA levels in the RNA-seq data. However, this does not speak to the activity of DNA repair proteins that could be altered by acetylation regardless of their level of expression. Therefore, we would argue that this is still a plausible hypothesis, also indirectly supported in published studies as we indicated. As described above, we have rewritten the manuscript to provide more focus on the RNA-seq component. However, we would like include this speculative part of the discussion as it raises testable hypotheses for future studies. We have made some changes to the text to minimize this section and to clarify the distinction between hypotheses related DNA repair gene expression level and protein activity.

We have eliminated panel D from Figure 6, as this method of quantification, while of interest generally as a method, did not change the conclusions, and therefore seemed unnecessary with the goal of restructuring the manuscript and reducing some of the instability focus.

The inclusion of MSH2 in the Materials and methods was a copy-paste error from our previous study and has been removed.

3) The methodology regarding how the mice colonies were made and maintained is confusing. Are these mice heterozygous or homozygous for Q111? Perhaps a supplementary breeding scheme diagram? This is vital for data interpretation. Also, please include the Q111 genotype in the main text as well as in the Materials and methods.

We have modified Figure 1 with a schematic to broadly illustrate the crosses. This now better sets up the experiment, together with the immunohistochemistry for *Hdac2* and *Hdac3* KO. In remaking this figure, we moved a part of it with data specific for the *Hdac2* KO to a new supplementary figure (Figure 1—figure supplement 2). We also made a new supplementary figure (Figure 1—figure supplement 1) detailing the crosses.

We have also simplified the nomenclature of the mice, using *Hdac2/3* KO and *Hdac2/3* WT rather than *Hdac2/3*^Δ/Δ^ and *Hdac2/3*^+/+^. The simplified nomenclature represents the combination of *Hdac2/3* and D9-Cre genotypes (as explained in the Materials and methods and shown in Figure 1—figure supplement 1) and is therefore a better way to represent the loss of expression or wild-type expression of HDAC2/3.

All mice are heterozygous for the *Htt*^Q111^ allele and we have clarified this throughout the text.

4) Presentation of RNAseq data is difficult to follow and not fully described. For instance, it is not clear from the text whether genes that are consistently altered in HD are restored in expression and information on statistical enrichment. Similarly, the comparison with Langfelder et al., data is unclear. Suggest to use graphical visuals representing gene changes and observed overlaps instead of tables or guide readers' eye to specific rows and columns being discussed when referencing figures. Further, suggest adding additional figure references in the fourth paragraph of the subsection “Impact of MSN-specific Hdac2 knockout on the transcriptome” for RNAseq comparisons to make it more clear so to which datasets text is referring to. Finally, conclusions relating to analysis from RNAseq does not appear to be consistent with the data. While the manuscript states that none of the rescued genes met the adjusted p-value cut-off (subsection “A subset of genes dysregulated in Htt^Q111^ striata is reversed by knockout of Hdac2 in MSNs”), the author's interpretation of results suggests a rescue effect (subsection “A subset of genes dysregulated in Htt^Q111^ striata is reversed by knockout of Hdac2 in MSNs”, last paragraph). Due to the lack of significance, would recommend re-phrasing to a suggested rescue rather than a definite one.

We have made several changes to better display the data and/or to make them more accessible

– For the number of DEGs (Figure 2B), we now display the data as graphs rather than tables. For the overlaps between the DEGs and enriched pathways between the different 2-way contrasts, we removed the overlap tables and created a new figure (Figure 3—figure supplement 1) in order to make these tables more visible. In the Results section we refer to the highlighted cells within these tables so that the data can be readily located. We have also defined each of the four 2-way contrasts as contrast 1, 2, 3 and 4 (labeled in Figure 2A), allowing us to refer clearly to these throughout.

– We have incorporated heat maps in Figure 4 and in Figure 2—figure supplement 4 to better display the data

Rescue of *Htt*^Q111^-dysregulated genes: We have redone these analyses to ask the question in a different and more straightforward way. The way this question was initially addressed was based on relative enrichment of gene dysregulation by *Hdac2* KO in *Htt*^Q111^ compared to WT with the aim of subtracting a “background effect” of the *Hdac2* KO. However, this does not necessarily allow us to test directly for normalization of absolute gene expression levels.

We have now performed the following analyses to more directly address the question of gene expression rescue:

1) We identified genes whose expression levels are not significantly different between wild-type mice and *Htt*^Q111^ mice harboring the *Hdac2* KO mutation. The results of this analyses are shown in Figure 4. and comprised ~12% of all *Htt*^Q111^-dysregulated genes.

2) As the reviewers point out, relatively few of the *Htt*^Q111^-dysregulated genes met a p value significance threshold corrected for multiple testing. We therefore analyzed the set of “common dysregulated genes” determined above in the intersection with the Langfelder data to determine whether these were corrected. These results are shown in Figure 4—figure supplement 1.

3) It was apparent from the “common dysregulated genes” that, regardless of the absolute effect of the *Hdac2* KO on gene expression, the presence of the KO mutation lessened the impact of the *Htt*^Q111^ allele relative to the *Htt*^+/+^ allele. This is essentially what our original analysis had captured, which we focus on now only in the subset of significant “common dysregulated genes”. This is an interesting observation that we have highlighted in Figure 5.

Lack of statistical significance: We agree that caution is needed with interpretation. In the Results we have used the term “possible rescue” in line with the reviewers’ recommendation. In the Discussion, we also included a sentence highlighting this caveat (Discussion, fourth paragraph). We have now also placed more focus on the significant genes that overlapped with the Langfelder significant genes.

5) While the Discussion addresses some of the previous studies investigating HDACs and HDAC inhibitors in HD, this is by no means comprehensive. Studies by Hoshino et al., 2003, and Qunti et al., 2010, should be included discussing HDAC expression in HD human and mouse tissue, respectively. Additionally, work by Jia et al., 2016, Chopra et al., 2016, and Siebzehnrubl et al., 2018, discuss the effects of various HDAC inhibitors on models of HD. Please include these in your Discussion section and also give a brief introduction to this previous work in the Introduction.

Thank you for pointing out these omissions. We have now included discussion of Jia et al., 2016, Chopra et al., 2016, and Siebzehnrubl et al., 2018, as well as additional studies of HDAC inhibitors in mice in the Discussion in relation to transcriptional dysregulation and nuclear pathology, and included these studies in the Introduction. The work from Hoshino et al., 2003, and Quinti et al., 2010, was not easily incorporated into our Discussion as this detracted from the major points. Therefore, we mentioned these papers in the Results (subsection “Impact of MSN-specific *Hdac2* knockout and of the *Htt*^Q111^ allele on the transcriptome”, first paragraph) in relation to the observation that the *Htt*^Q111^ allele did not alter *Hdac* expression, as these studies are directly relevant to this point.